# From Misclassifications to Outliers: Joint Reliability Assessment in Classification

## Abstract

Building reliable classifiers is a fundamental challenge for deploying machine learning in real-world applications. A reliable system should not only detect out-of-distribution (OOD) inputs but also anticipate in-distribution (ID) errors by assigning low confidence to potentially misclassified samples. Yet, most prior work treats OOD detection and failure prediction as separated problems, overlooking their closed connection. We argue that reliability requires evaluating them jointly. To this end, we propose a unified evaluation framework that integrates OOD detection and failure prediction, quantified by our new metrics DS-F1 and DS-AURC, where DS denotes double scoring functions. Experiments on the OpenOOD benchmark show that double scoring functions yield classifiers that are substantially more reliable than traditional single scoring approaches. Our analysis further reveals that OOD-based approaches provide notable gains under simple or far-OOD shifts, but only marginal benefits under more challenging near-OOD conditions. Beyond evaluation, we extend the reliable classifier SURE and introduce SURE+, a new approach that significantly improves reliability across diverse scenarios. Together, our framework, metrics, and method establish a new benchmark for trustworthy classification and offer practical guidance for deploying robust models in real-world settings. Code will be released upon publication.

## 1 Introduction

Deploying machine learning classifiers in safety-critical domains such as fire and smoke detection demands model robustness beyond high benchmark accuracy. In real-world environments, a reliable system must not only detect actual fire and smoke, but also avoid false alarms caused by visually similar yet harmless phenomena (e.g., fog, steam, or unusual lighting). Equally important, the system should be able to recognize its own uncertainty and refrain from making overconfident misclassifications. Failures in either direction–failing to detect a true fire or generating frequent false alarms–can lead to severe consequences, ranging from safety risks to a loss of user trust. Ensuring reliability under these conditions is therefore a fundamental and urgent challenge.

The research community has actively explored two directions that are highly related to reliability. The first is out-of-distribution (OOD) detection (Yang et al., 2022; Zhang et al., 2023; Yang et al., 2024), which aims to identify inputs that deviate from the training distribution and should not be trusted. The second is failure prediction (Corbière et al., 2019; Zhu et al., 2022), which estimates whether a classifier's prediction on an in-distribution (ID) sample is correct. Both directions have established benchmarks, specialized algorithms, and thriving research communities. However, most existing works study these two aspects separately, treating them as independent problems.

In real-world applications, a classifier must handle both in-distribution (ID) and out-of-distribution (OOD) inputs. This creates the challenge of jointly addressing OOD detection and failure prediction, so that model reliability can be assessed in a unified way. Suppose the model produces two scores for each input: *i) an OOD detection score $s_{\text{OOD}}$, and ii) an in-distribution confidence score $s_{\text{ID}}$.* Together, these scores form a binary decision system:

- First, we ask: Is the input ID or OOD? If $s_{\text{OOD}}$ exceeds a threshold $\tau_{\text{OOD}}$, the input is accepted as ID.

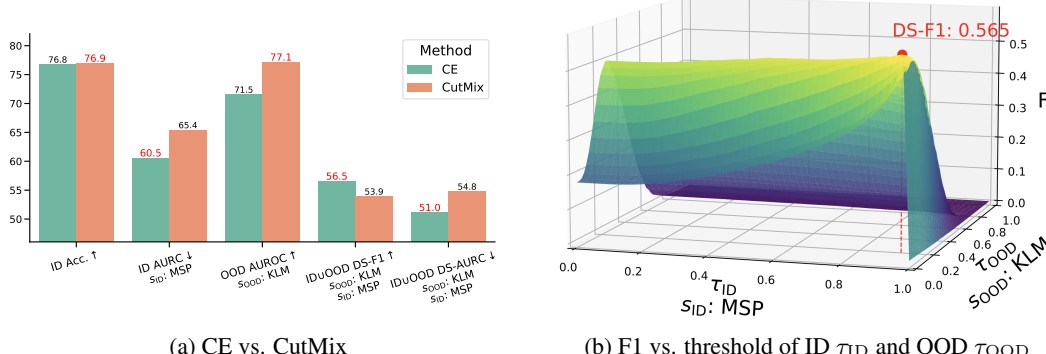

(a) CE vs. CutMix

(b) F1 vs. threshold of ID $\tau_{\mathrm{ID}}$ and OOD $\tau_{\mathrm{OOD}}$

Figure 1: **Joint evaluation of ID and OOD.** a) ResNet-18 models trained with cross-entropy (CE) and CutMix on CIFAR-100, evaluated on CIFAR-100 (ID) and MNIST (OOD). b) F1 score surface of CE showing how DS-F1 jointly considers ID and OOD scores to find the optimal classification performance.

- Next, if the input is ID, we ask: Can we trust the predicted label? This is determined by applying another threshold $\tau_{\mathrm{ID}}$ on $s_{\mathrm{ID}}$.

Addressing both dimensions at once is challenging. A related direction is selective classification Geifman & El-Yaniv (2017), which evaluates both ID accuracy and OOD rejection within a unified framework Xia & Bouganis (2022). However, most selective classification methods rely on a **single scoring function** with one decision threshold to accept or reject predictions. This approach does not fully take advantage of the specialized progress made in the OOD detection and failure prediction communities.

The challenge of evaluation has already been emphasized in prior studies. For instance, the OpenOOD benchmark Yang et al. (2022); Zhang et al. (2023) shows that no single method consistently excels across all metrics, complicating the choice of a training strategy or post-hoc approach for practical use. While this issue arises in the context of OOD detection alone, it becomes even more complex when ID accuracy is also considered, since OOD detection and failure prediction are typically evaluated in isolation. Such fragmented evaluations lead to inconsistencies and hinder reliable deployment. What is needed instead is a unified metric that more intuitively captures performance on the joint task.

To illustrate the tension between ID reliability and OOD performance, we train two ResNet-18 He et al. (2016) classifiers on CIFAR-100 Krizhevsky et al. (2009): one using standard cross-entropy loss (CE) and the other with CutMix Yun et al. (2019). We then evaluate both models on CIFAR-100 (ID) and MNIST (OOD) LeCun (1998). As shown in the first three columns of Figure 1a, CutMix slightly improves ID accuracy and boosts OOD detection performance according to AUROC metric and using KLM Basart et al. (2022) as the scoring function for $s_{\mathrm{OOD}}$, but it decreases reliability on ID samples according to AURC metric on the ID test set when taking MSP Hendrycks & Gimpel (2017) for $s_{\mathrm{ID}}$. This highlights a common dilemma: both ID and OOD performance are crucial for building a reliable classifier in real-world settings. Focusing on only ID accuracy or OOD detection can be misleading, making it difficult to determine which model is truly dependable.

To address this challenge, we propose a unified framework that treats OOD detection and failure prediction as complementary aspects of classifier reliability. Rather than evaluating these tasks separately, we assess them jointly using two scoring functions and their corresponding thresholds: $\tau_{\mathrm{OOD}}$ for detecting OOD samples and $\tau_{\mathrm{ID}}$ for assessing ID classification confidence. This double-scoring approach divides all samples into four categories: *True Accept*, *True Reject*, *False Accept*, and *False Reject*. For example, True Accept refers to ID samples that pass both thresholds and are correctly classified by the model. Using these categories, we extend standard metrics such as AURC and F1 to account for both ID and OOD data, resulting in DS-AURC and DS-F1 (DS = Double Scoring). For instance, DS-F1 searches for the best F1 score across the double-scoring surface, as illustrated in Figure 1b. These metrics give a more accurate measure of a classifier's reliability and help identify truly robust models, avoiding misleading conclusions that arise when

considering only one aspect in isolation. As shown in the last two columns of Figure 1a, when considering simultaneously KLM and MSP as $s_{\text{OOD}}$ and $s_{\text{ID}}$, our DS-F1 and DS-AURC metrics show that CE achieves a higher DS-F1 and a lower DS-AURC, indicating better overall performance when evaluating ID and OOD performance together.

We show that for double scoring functions, which jointly consider the OOD score ($s_{\text{OOD}}$) and the ID confidence ($s_{\text{ID}}$), the evaluation metrics naturally reduce to their single-score counterparts when only one score is used. Concretely, DS-AURC is lower bounded by the standard AURC, while DS-F1 is guaranteed to be no worse than the best F1 obtained from a single scoring function. This ensures that double scoring at least matches single-score methods while providing a more faithful measure of classifier reliability. To validate this, we conduct extensive experiments on OpenOOD Yang et al. (2022), showing that double scoring consistently produces more robust and reliable classifiers. Interestingly, post-hoc OOD scores ($s_{\text{OOD}}$) complement MSP ($s_{\text{ID}}$) effectively under far-OOD shifts, yet provide little additional benefit under near-OOD conditions, highlighting the limits of relying solely on advanced post-hoc OOD detection approaches.

Beyond the unified evaluation metric, we extend the reliable classifier SURE Li et al. (2024b), originally focused on failure prediction, to handle both ID and OOD scenarios. Building on this, we introduce SURE+, a streamlined and more powerful version that incorporates recent advances in both OOD detection and failure prediction. With this unified and simplified design, SURE+ achieves significantly higher reliability when evaluated on both ID and OOD samples.

Our key contributions are as follows:

- We reveal that *OOD detection* and *failure prediction*, though often studied in isolation, are inherently complementary aspects of classifier reliability. Evaluating them separately can lead to misleading conclusions, whereas a unified perspective provides a more faithful reflection of real-world deployment needs.

- To this end, we introduce two complementary metrics, **DS-F1** and **DS-AURC**, which jointly measure a classifier's ability to detect OOD inputs and anticipate its own misclassifications. These metrics offer a principled and comprehensive way to evaluate model reliability.

- Extensive experiments on the OpenOOD benchmark demonstrate that our unified framework consistently identifies classifiers that are substantially more robust and trustworthy than those optimized for OOD detection or failure prediction alone.

- Beyond evaluation, we propose **SURE+**, an improved reliable classifier that builds on SURE by integrating recent advances in both OOD detection and failure prediction. Experiments show that SURE+ achieves state-of-the-art reliability across diverse scenarios.

Code will be released upon publication.

## 2 RELATED WORK

**Out-of-distribution detection.** Out-of-distribution (OOD) detection has been widely studied for enhancing the robustness and reliability of models. Existing methods can be broadly divided into post-hoc and training-based methods. Post-hoc methods, such as MSP Hendrycks & Gimpel (2017), Energy Liu et al. (2020), GradNorm Huang et al. (2021), aim to leverage signals already contained in a trained model to distinguish OOD from ID inputs, whether from the model's logits, intermediate features, or their gradients. The training-based methods explicitly modify the learning process, for example, by incorporating auxiliary outliers Hendrycks et al. (2019); Zhu et al. (2023a), masking ID inputs Li et al. (2023), or applying regularization losses Ming et al. (2023), to make models more sensitive to OOD data. Despite their differences, these approaches share the common goal of providing a reliable scoring function, which, combined with a threshold, filters out OOD inputs and prevents the system from mistakenly trusting predictions on them.

**Failure prediction and selective classification.** Beyond detecting distributional novelty, a key challenge is to predict whether a model's output is correct. Post-hoc scoring functions introduced for OOD detection can also be applied to failure prediction, e.g. MSP Hendrycks & Gimpel (2017),

while some works exploit intermediate representations of a trained model and train auxiliary modules for estimating prediction errors Corbière et al. (2019); Luo et al. (2021); Yu et al. (2021); Shen et al. (2023). SIRC Xia & Bouganis (2022) adopts a selective classification perspective Kim et al. (2023); Geifman & El-Yaniv (2017), combining multiple post-hoc scores to filter unreliable predictions and better separate system failures from reliable outputs. However, its evaluation still treats ID and OOD data separately, despite the method's aim to address both. Evaluating a classification system under joint ID and OOD inputs is more realistic but under-explored, as current metrics largely consider these dimensions independently, motivating the need for unified system-level evaluation.

**Training strategies for reliable classifiers.** Post-hoc scoring methods rely on pre-trained models, and their effectiveness is closely tied to the underlying training scheme. Data augmentation strategies such as Mixup Thulasidasan et al. (2019) and RegMixup Pinto et al. (2022) enhance generalization and OOD detection. Optimization techniques like SAM Foret et al. (2021) and F-SAM Li et al. (2024a) encourage flat minima, while weight averaging approaches like SWA Izmailov et al. (2018) and SWAG Maddox et al. (2019) improve stability and calibration. More recent pipelines such as FMFP Zhu et al. (2023b) and SURE Li et al. (2024b) integrate these ideas, achieving strong performance in failure prediction and robustness. Yet, the behavior of SURE under OOD exposure remains underexplored. In this work, we introduce SURE+, an enhanced version designed to improve reliability and robustness in realistic scenarios where both ID and OOD samples are present.

**Existing evaluation metrics for classification system.** Metrics for failure prediction and OOD detection are typically based on the selective classification principle, where a scoring function and threshold decide whether to trust a prediction. They can be grouped into single-threshold metrics, such as AUROC, AUPR (Hendrycks & Gimpel, 2017; Yang et al., 2022; Zhang et al., 2023), FPR@95, and F1 score, which evaluate performance at a fixed threshold, and multi-threshold metrics, such as AURC Geifman et al. (2019) and recently proposed AUGRC Traub et al. (2024), which aggregate performance across thresholds to assess risk over coverage.

However, these metrics implicitly assume that a single scoring rule suffices for building a reliable system. In practice, a trustworthy classifier must both distinguish OOD from ID samples and further separate correctly classified ID instances from other accepted predictions. This calls for a more comprehensive evaluation paradigm, where two scoring functions with corresponding thresholds are more effective than a single one. To this end, we propose DS-F1 and DS-AURC, which naturally extend selective classification by incorporating double scoring to better capture system reliability.

## 3 DS-F1 and DS-AURC: Novel Evaluation Metrics

We here consider that double scoring functions perform OOD detection and failure prediction, each has its own threshold, and the functions could share the same post-hoc method. This matches more in real-world settings, for instance, one can use MSP score Hendrycks & Gimpel (2017) for both OOD and failure prediction, but can also apply Energy score Liu et al. (2020) for OOD detection and MSP score for failure prediction. This motivates us to extend the existing single-rule framework into a double-rule setting, leading to our proposed DS-F1 and DS-AURC metrics.

Let $\mathcal{D} = \mathcal{D}_{\text{ID}} \cup \mathcal{D}_{\text{OOD}}$ be the evaluation dataset, comprising both ID and OOD samples, with $N$ samples in total. For an input $x$, a model $m$ produces a prediction $m(x)$. We then have two scoring functions based on $m$:

1. An **OOD detection score** $s_{\text{OOD}}(x) \in \mathbb{R}$, where higher values indicate a higher likelihood of the sample being ID.
2. A **failure prediction score** $s_{\text{ID}}(x) \in \mathbb{R}$, where higher values indicate higher likelihood in the prediction $m(x)$ being correct.

The decision to accept a prediction is based on two corresponding thresholds, $\tau_{\text{OOD}}$ and $\tau_{\text{ID}} \in \mathbb{R}$.

### 3.1 DS-F1

The classical F1-score balances precision and recall for a single binary decision rule. We extend this concept to our double-scoring setting to find the optimal joint operating point of the system. The goal

is to **accept correctly classified ID samples** while rejecting both OOD samples and misclassified ID samples.

Specifically, for any threshold pair $(\tau_{\text{OOD}}, \tau_{\text{ID}})$, the acceptance set is

$$\mathcal{A}(\tau_g, \tau_h) = \{i : s_{\text{OOD}}(x_i) \geq \tau_{\text{OOD}} \wedge s_{\text{ID}}(x_i) \geq \tau_{\text{ID}}\}. \tag{1}$$

To compute the F1-score and avoid confusion with standard "true positive / false negative" terminology, we adopt an acceptance-oriented notation. We define:

- **True Accept (TA):** ID samples that are accepted and correctly classified.

$$\text{TA}(\tau_{\text{OOD}}, \tau_{\text{ID}}) = \sum_{i \in \mathcal{D}_{\text{ID}}} \mathbb{I}(i \in \mathcal{A}(\tau_{\text{OOD}}, \tau_{\text{ID}})) \cdot \mathbb{I}(m(x_i) = y_i)$$

- **False Accept (FA):** Samples that are accepted but not correct, including accepted OOD samples and misclassified ID samples.

$$\text{FA}(\tau_{\text{OOD}}, \tau_{\text{ID}}) = \underbrace{\sum_{i \in \mathcal{D}_{\text{OOD}}} \mathbb{I}(i \in \mathcal{A}(\tau_{\text{OOD}}, \tau_{\text{ID}}))}_{\text{FA from accepted OOD}} + \underbrace{\sum_{i \in \mathcal{D}_{\text{ID}}} \mathbb{I}(i \in \mathcal{A}(\tau_{\text{OOD}}, \tau_{\text{ID}})) \cdot \mathbb{I}(m(x_i) \neq y_i)}_{\text{FA from accepted misclassified ID}}$$

- **False Reject (FR):** ID samples that are not correctly accepted, either because they are rejected or misclassified when accepted.

$$\text{FR}(\tau_{\text{OOD}}, \tau_{\text{ID}}) = \sum_{i \in \mathcal{D}_{\text{ID}}} \left[ \underbrace{\mathbb{I}(i \notin \mathcal{A}(\tau_{\text{OOD}}, \tau_{\text{ID}}))}_{\text{Rejected ID}} + \underbrace{\mathbb{I}(i \in \mathcal{A}(\tau_{\text{OOD}}, \tau_{\text{ID}})) \cdot \mathbb{I}(m(x_i) \neq y_i)}_{\text{Misclassified ID}} \right]$$

Note that misclassified but accepted ID samples are counted in both FA and FR: they are ID but not correctly predicted (hence FR), and are also wrongly accepted (hence FA). This overlap is inherent in the double-scoring system, but does not lead to inconsistency. Indeed, by construction, we have

$$\text{TA} + \text{FR} = |\mathcal{D}_{\text{ID}}|, \qquad \text{TA} + \text{FA} = |\mathcal{A}(\tau_{\text{OOD}}, \tau_{\text{ID}})|,$$

so the resulting definitions of precision and recall are equivalent to their standard forms and can be expressed more compactly as

$$\text{Precision}(\tau_{\text{OOD}}, \tau_{\text{ID}}) = \frac{\text{TA}(\tau_{\text{OOD}}, \tau_{\text{ID}})}{|\mathcal{A}(\tau_{\text{OOD}}, \tau_{\text{ID}})|}, \quad \text{Recall}(\tau_{\text{OOD}}, \tau_{\text{ID}}) = \frac{\text{TA}(\tau_{\text{OOD}}, \tau_{\text{ID}})}{|\mathcal{D}_{\text{ID}}|}.$$

The **DS-F1** is then defined as the maximum achievable F1-score across all possible threshold pairs, capturing the best possible system performance:

$$\text{DS-F1} = \max_{\tau_{\text{OOD}}, \tau_{\text{ID}}} \left( \frac{2 \cdot \text{Precision}(\tau_{\text{OOD}}, \tau_{\text{ID}}) \cdot \text{Recall}(\tau_{\text{OOD}}, \tau_{\text{ID}})}{\text{Precision}(\tau_{\text{OOD}}, \tau_{\text{ID}}) + \text{Recall}(\tau_{\text{OOD}}, \tau_{\text{ID}})} \right) \tag{2}$$

This formulation directly captures the best achievable joint operating point when both scoring functions are applied simultaneously, and serves as a point-wise evaluation metric complementary to area-based ones. As shown in Figure 1b, unlike the classical way to achieve the F1 score, which searches over a single threshold each time according to a single scoring method, DS-F1 searches over threshold pairs, thereby generalizing the decision boundary to a two-dimensional space.

## 3.2 DS-AURC

While DS-F1 identifies the best operating point, we also need to evaluate the system's performance across the entire range of possible operating points. To achieve this, we generalize the AURC metric and extend the notion of multi-threshold evaluation to the double-scoring setting.

**Coverage.** Given an acceptance rule induced by a threshold $\tau$ of a scoring function $s$, according to Eq. 1, the coverage $u$ is defined as the proportion of accepted samples ID samples by the system:

$$u = |\mathcal{A}(\tau) \cap \mathcal{D}_{\text{ID}}| / |\mathcal{D}_{\text{ID}}|, \tag{3}$$

This definition is consistent with the ID-only case, since when $\mathcal{D}_{\text{OOD}} = \varnothing$, Eq. 3 reduces to the standard coverage definition in selective classification.

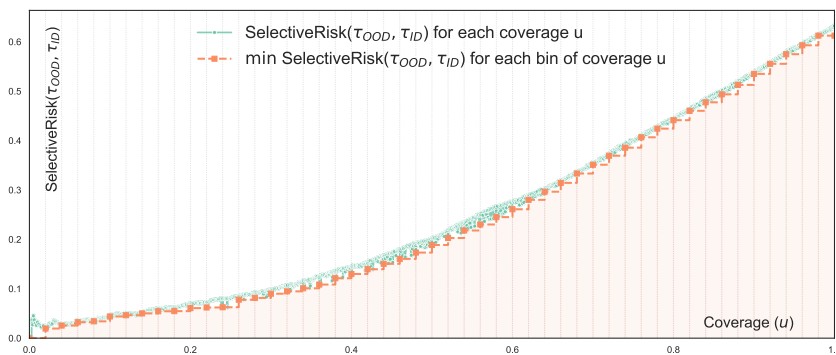

Figure 2: **Example of DS-AURC.** Selective risk versus coverage for double scoring is shown in green. DS-AURC uses the best risk in each coverage bin (orange, in Eq. 7), resulting in a lower (better) AURC. Results with ResNet-18 on CIFAR-100 (ID) and CIFAR-10 (OOD), using VIM Wang et al. (2022) for the OOD score and MSP Hendrycks & Gimpel (2017) for the ID score.

**AURC on ID-only test set.** Let $Z = \mathbb{I}(m(x) \neq y)$ denote the indicator for misclassification. For an evaluation set containing only ID data, the selective risk at threshold $\tau$ and the resulting AURC are defined as

$$\text{SelectiveRisk}(\tau) = \frac{\sum_{i \in \mathcal{D}_{\text{ID}}} Z_i \cdot \mathbb{I}(i \in \mathcal{A}(\tau))}{|\mathcal{A}(\tau)|}, \quad \text{AURC} = \int_0^1 \text{SelectiveRisk}(\tau(u)) \, du, \quad (4)$$

where the ratio is defined as 0 when $|\mathcal{A}(\tau)| = 0$, and $\tau(u)$ is the unique threshold corresponding to coverage $u$ under a single scoring function.

**AURC on ID+OOD evaluation set.** When we consider a more realistic case, the evaluation set contains both ID and OOD data, and $\mathcal{A}(\tau)$ may include OOD samples. In this case, the selective risk must be redefined to reflect the accuracy of the ID portion of accepted samples:

$$\text{SelectiveRisk}(\tau) = \frac{\sum_{i \in \mathcal{D}_{\text{ID}}} Z_i \cdot \mathbb{I}(i \in \mathcal{A}(\tau)) \; + \; |\mathcal{A}(\tau) \cap \mathcal{D}_{\text{OOD}}|}{|\mathcal{A}(\tau)|} \quad (5)$$

This definition treats not only accepted misclassified ID as risks but also each accepted OOD sample, since once the system accepts a prediction given by an OOD input, it indicates a failure. Note that when $\mathcal{D}_{\text{OOD}} = \varnothing$, Eq. 5 reduces to the selective risk in Eq. 4, so this definition is compatible with the ID-only testing case. Meanwhile, the calculation of AURC will be the same as in Eq. 4.

**DS-AURC.** We still consider the ID+OOD evaluation setting here. When two scoring functions $s_{\text{OOD}}$ and $s_{\text{ID}}$ are jointly applied, a given coverage level $u$ may correspond to multiple threshold pairs $(\tau_{\text{OOD}}, \tau_{\text{ID}})$, each inducing a distinct acceptance set $\mathcal{A}(\tau_{\text{OOD}}, \tau_{\text{ID}})$. We define the selective risk for a threshold pair similar to Eq. 5 as follows:

$$\text{SelectiveRisk}(\tau_{\text{OOD}}, \tau_{\text{ID}}) = \frac{\sum_{i \in \mathcal{D}_{\text{ID}}} Z_i \cdot \mathbb{I}(i \in \mathcal{A}(\tau_{\text{OOD}}, \tau_{\text{ID}})) \; + \; |\mathcal{A}(\tau_{\text{OOD}}, \tau_{\text{ID}}) \cap \mathcal{D}_{\text{OOD}}|}{|\mathcal{A}(\tau_{\text{OOD}}, \tau_{\text{ID}})|} \quad (6)$$

Since at a fixed coverage $u$ there may exist multiple threshold pairs, the selective risk is no longer unique but a set of values. With Eq. 6, we therefore define the DS-AURC as

$$\text{DS-AURC} = \int_0^1 \phi\left\{\text{SelectiveRisk}(\tau_{\text{OOD}}, \tau_{\text{ID}}) : (\tau_{\text{OOD}}, \tau_{\text{ID}}) \in \mathbb{R}^2, \; \frac{|\mathcal{A}(\tau_{\text{OOD}}, \tau_{\text{ID}}) \cap \mathcal{D}_{\text{ID}}|}{|\mathcal{D}_{\text{ID}}|} = u\right\} du \quad (7)$$

where $\phi$ is an aggregation operator over risks associated with the same coverage level, and the definition of the coverage here is also consistent with Eq. 3. In practice, AUC is often computed via the trapezoidal rule, which by default takes the last item when multiple risks share the same coverage. We instead set $\phi = \min$ to capture the best achievable risk at each coverage level. Consequently, the DS-AURC reflects the smallest possible coverage–risk area, highlighting the optimistic bound of model performance under selective prediction.

| Method | Model: ResNet18 Training/Validation set: CIFAR100 training set Evaluation set: CIFAR100 test set + Near/Far OOD sets | | | | Model: DeiT-B Training/Validation set: ImageNet1K training set Evaluation set: ImageNet1K test set + Near/Far OOD sets | | | |
| | Single Scoring | | Double Scoring | | Single Scoring | | Double Scoring | |
| | F1 ↑ | AURC ↓ | DS-F1 ↑ | DS-AURC ↓ | F1 ↑ | AURC ↓ | DS-F1 ↑ | DS-AURC ↓ |
| MSP Hendrycks & Gimpel (2017) | 67.42 / 57.03 | 202.59 / 368.07 | 67.42 / 57.03 | 202.38 / 367.56 | 70.55 / 79.05 | 241.44 / 98.15 | 70.55 / 79.05 | 241.35 / 98.13 |
| OpenMax Bendale & Boult (2016) | 64.96 / 55.41 | 264.21 / 444.53 | 67.42 / 57.09 | 201.78 / 365.87 | 67.06 / 77.73 | 404.98 / 248.49 | 70.70 / 79.60 | 239.42 / 93.32 |
| ODIN Liang et al. (2018) | 65.89 / 56.54 | 219.63 / 358.68 | 67.42 / 58.75 | 199.82 / 330.49 | 67.48 / 76.21 | 322.35 / 192.30 | 70.55 / 79.05 | 239.54 / 97.16 |
| MDS Lee et al. (2018) | 56.19 / 47.82 | 339.54 / 464.67 | 67.43 / 57.22 | 201.78 / 355.07 | 68.60 / 78.12 | 252.45 / 99.23 | 70.56 / 79.26 | 226.20 / 77.30 |
| Gram Hendrycks et al. (2021a) | 53.23 / 46.09 | 536.12 / 548.61 | 67.42 / 59.16 | 201.14 / 313.08 | 64.87 / 73.50 | 488.21 / 424.83 | 70.55 / 79.05 | 240.89 / 97.43 |
| EBO Liu et al. (2020) | 66.62 / 57.41 | 205.79 / 345.83 | 67.42 / 57.76 | 197.99 / 342.04 | 66.92 / 76.20 | 355.94 / 214.35 | 70.55 / 79.05 | 240.78 / 97.78 |
| GradNorm Huang et al. (2021) | 63.31 / 47.95 | 322.48 / 518.60 | 67.45 / 57.05 | 201.08 / 365.74 | 64.87 / 73.51 | 532.10 / 454.47 | 70.55 / 79.05 | 240.78 / 97.78 |
| ReAct Sun et al. (2021) | 66.32 / 58.03 | 213.53 / 335.99 | 67.43 / 58.44 | 199.61 / 331.40 | 67.99 / 77.67 | 290.84 / 131.86 | 70.56 / 79.05 | 239.29 / 93.36 |
| MLS Basart et al. (2022) | 66.90 / 57.47 | 203.87 / 346.30 | 67.42 / 57.71 | 197.76 / 343.24 | 69.28 / 78.36 | 303.42 / 157.43 | 70.55 / 79.05 | 240.66 / 98.10 |
| KLM Basart et al. (2022) | 65.52 / 48.90 | 306.23 / 524.84 | 67.44 / 57.09 | 199.23 / 351.05 | 70.54 / 79.37 | 249.44 / 93.00 | 70.63 / 79.41 | 229.50 / 83.42 |
| VIM Wang et al. (2022) | 59.91 / 55.07 | 270.50 / 359.40 | 67.46 / 58.94 | 198.39 / 321.09 | 68.21 / 77.92 | 299.90 / 139.83 | 70.55 / 79.18 | 231.73 / 81.86 |
| KNN Sun et al. (2022) | 67.10 / 57.47 | 220.78 / 347.05 | 67.67 / 58.26 | 197.30 / 334.68 | 67.91 / 77.81 | 261.15 / 102.68 | 70.55 / 79.10 | 229.33 / 79.05 |
| DICE Sun & Li (2022) | 65.82 / 57.11 | 214.11 / 347.94 | 67.43 / 58.04 | 198.75 / 338.83 | 64.96 / 74.14 | 464.09 / 313.53 | 70.55 / 79.06 | 239.46 / 96.42 |
| SIRC(MSP,$\|z\|_1$) Xia & Bouganis (2022) | 67.36 / 56.96 | 199.20 / 354.96 | 67.44 / 57.05 | 197.08 / 353.41 | 70.43 / 78.87 | 244.86 / 102.46 | 70.55 / 79.05 | 240.78 / 97.78 |
| SIRC(MSP,Res.) Xia & Bouganis (2022) | 67.33 / 57.00 | 198.86 / 354.72 | 67.42 / 57.03 | 197.21 / 353.08 | 70.42 / 78.86 | 245.16 / 102.87 | 70.55 / 79.05 | 241.04 / 98.09 |
| SIRC(-H,$\|z\|_1$) Xia & Bouganis (2022) | 67.24 / 57.18 | 198.46 / 349.53 | 67.44 / 57.25 | 196.46 / 348.47 | 69.74 / 78.80 | 264.08 / 118.81 | 70.55 / 79.05 | 241.06 / 98.08 |
| SIRC(-H,Res.) Xia & Bouganis (2022) | 67.20 / 57.20 | 198.66 / 349.68 | 67.42 / 57.26 | 196.60 / 348.36 | 69.74 / 78.80 | 267.67 / 118.81 | 70.55 / 79.05 | 241.04 / 98.05 |
| ID Acc. | 77.32 | | | | 81.79 | | | |

Table 1: **Experiments are conducted on CIFAR100 Krizhevsky et al. (2009) with ResNet-18 He et al. (2016), and on ImageNet Deng et al. (2009) with DeiT-B Touvron et al. (2021).** For double scoring metrics, we use MSP as the ID score ($s_{\mathrm{ID}}$) and apply different OOD scores ($s_{\mathrm{OOD}}$), reporting results on both Near- and Far-OOD tests. CIFAR-100 experiments are repeated three times, and average results are reported. The top five methods for each metric are highlighted with a color gradient from light blue to **dark blue**.

**Summary and properties of DS-F1 and DS-AURC.** The proposed DS-F1 and DS-AURC metrics provide a principled generalization of the classical single-threshold evaluation pipeline. Their detailed implementation, including pseudo-code for computing DS-F1 and DS-AURC, is provided in Appendix A.1.1. Meanwhile, as formally shown in Appendix A.1.2, DS-F1 is guaranteed to be at least as high as the standard F1, and DS-AURC is guaranteed to be lower than or equal to the standard AURC. This ensures that adopting double scoring never worsens evaluation outcomes.

These metrics exhibit several notable properties. First, they naturally generalize the traditional metrics: when one threshold is fixed, DS-F1 and DS-AURC reduce to the standard F1 and AURC, maintaining full consistency with previous pipelines. Second, our metrics provide clear performance guarantees: DS-F1 is always greater than or equal to F1, and DS-AURC is always less than or equal to AURC. Third, double scoring can change the ranking of methods. For example, a model with the lowest single-score AURC may not achieve the lowest DS-AURC, since DS-AURC evaluates multiple threshold combinations at each coverage level. As shown in Figure 2, for each coverage bin, DS-AURC selects the minimal risk (orange points), resulting in a risk surface that is always equal to or better than that of single scoring.

Taken together, DS-F1 and DS-AURC provide complementary perspectives: single-threshold (best operating point) and multi-threshold (global trade-off), thus offering a more faithful evaluation when OOD detection and failure prediction should be jointly considered.

## 4    SURE+: A TRAINING RECIPE FOR STRONGER CLASSIFICATION BASELINE

In this section, we propose an improved training recipe to build a stronger classification baseline. Our method builds on SURE Li et al. (2024b), which performs well on failure prediction but struggles with OOD detection. We revisit its pipeline and introduce key modifications: (1) remove CRL loss Moon et al. (2020) and replace the cosine similarity classification head (CSC) with a simple linear classifier to simplify training, (2) replace SWA Izmailov et al. (2018) with EMA for easier and more stable optimization, (3) introduce *RegPixMix*, a new augmentation adopted from PixMix Hendrycks et al. (2022) that complements RegMixup Pinto et al. (2022) to boost robustness, and (4) adopt F-SAM Li et al. (2024a) for better accuracy and reliability. Together, these changes yield consistent improvements across both ID and OOD evaluation.

Together, these modifications form a principled, unified training recipe that substantially improves both in-distribution (ID) and OOD performance. Building on SURE, we refer to this stronger, more reliable baseline as *SURE+*. We provide an introduction of SURE as well as the implementation details and hyperparameters of SURE+ to the Appendix A.5.

**Metric: DS-F1 ↑ ResNet-18 - Trained on CIFAR-100 training set. Evaluated on CIFAR100 test set + Near-OOD.**

| Training strategy | MSP | OpenMax | MDS | Gram | ReAct | KLM | VIM | KNN | SIRC | Acc. |
|---|---|---|---|---|---|---|---|---|---|---|
| Basic | 67.42 | 67.42 | 67.43 | 67.42 | 67.43 | 67.44 | 67.46 | 67.67 | 67.44 | 77.32 |
| Mixup | 67.86 | 67.98 | 67.87 | 67.87 | 67.87 | 67.88 | 67.87 | 68.06 | 67.87 | 78.47 |
| RegMixup | 68.77 | 68.77 | 68.77 | 68.77 | 68.78 | 68.78 | 68.77 | 68.81 | 68.78 | 79.35 |
| AugMix | 66.40 | 66.40 | 66.40 | 66.40 | 66.41 | 66.41 | 66.42 | 66.66 | 66.41 | 76.98 |
| PixMix | 66.65 | 66.65 | 66.65 | 66.65 | 66.78 | 66.68 | 66.69 | 66.84 | 66.67 | 77.20 |
| CutMix | 65.99 | 66.08 | 65.99 | 65.99 | 65.99 | 66.00 | 65.99 | 66.59 | 65.99 | 77.81 |
| SURE | 68.07 | 68.18 | 68.07 | 68.07 | 68.08 | 68.07 | 68.07 | 69.43 | 68.37 | 80.55 |
| - CSC | 68.72 | 68.72 | 68.72 | 68.73 | 69.05 | 68.85 | 68.74 | 68.76 | 68.72 | 80.36 |
| - CRL | 69.03 | 69.02 | 69.02 | 69.02 | 69.21 | 69.08 | 69.03 | 69.07 | 69.03 | 80.68 |
| + SWA→EMA | 69.24 | 69.24 | 69.24 | 69.24 | 69.45 | 69.29 | 69.26 | 69.27 | 69.24 | 80.54 |
| + SAM→F-SAM | 69.41 | 69.41 | 69.41 | 69.42 | 69.52 | 69.53 | 69.41 | 69.45 | 69.41 | 80.79 |
| + RegPixMix (**SURE+**) | **70.67** | **70.67** | **70.67** | **70.67** | **70.76** | **70.68** | **70.67** | **70.67** | **70.67** | **81.66** |

**Metric: DS-F1 ↑ ResNet-18 - Trained on CIFAR-100 training set. Evaluated on CIFAR100 test set + Far-OOD.**

| Training strategy | MSP | OpenMax | MDS | Gram | ReAct | KLM | VIM | KNN | SIRC | Acc. |
|---|---|---|---|---|---|---|---|---|---|---|
| Basic | 57.03 | 57.09 | 57.22 | 59.16 | 58.44 | 57.09 | 58.94 | 58.26 | 57.05 | 77.32 |
| Mixup | 58.02 | 61.29 | 58.34 | 58.04 | 58.03 | 58.18 | 58.53 | 59.72 | 58.02 | 78.47 |
| RegMixup | 55.08 | 58.16 | 55.41 | 56.44 | 55.83 | 55.15 | 55.60 | 58.22 | 55.11 | 79.35 |
| AugMix | 58.82 | 58.93 | 58.92 | 59.22 | 60.19 | 58.91 | 59.39 | 59.26 | 58.86 | 76.98 |
| PixMix | 57.06 | 57.04 | 57.05 | 60.38 | 58.83 | 57.40 | 58.13 | 57.53 | 57.53 | 77.20 |
| CutMix | 55.59 | 60.75 | 56.30 | 55.99 | 55.58 | 55.88 | 56.06 | 59.83 | 55.58 | 77.81 |
| SURE | 53.09 | 54.10 | 53.13 | 56.89 | 54.00 | 53.31 | 53.50 | 57.26 | 55.81 | 80.55 |
| - CSC | 56.17 | 58.58 | 56.94 | 57.04 | 60.45 | 58.27 | 60.16 | 60.49 | 56.20 | 80.36 |
| - CRL | 57.56 | 58.49 | 58.00 | 58.40 | 60.22 | 58.82 | **62.49** | 62.83 | 57.65 | 80.68 |
| + SWA→EMA | 57.34 | 58.24 | 57.77 | 57.99 | 60.71 | 58.40 | 60.01 | 62.48 | 57.36 | 80.54 |
| + SAM→F-SAM | 57.05 | 59.00 | 57.45 | 58.46 | 60.26 | 58.54 | 58.59 | 61.52 | 57.09 | 80.79 |
| + RegPixMix (**SURE+**) | **61.35** | **61.33** | **61.42** | **64.76** | **63.29** | **61.87** | 62.30 | **64.51** | **61.90** | **81.66** |

Table 2: **DS-F1 results on CIFAR-100 (ResNet-18) with different training strategies**. We report DS-F1 scores (higher is better) for both near- and far-OOD settings, along with ID accuracy. SURE+ consistently achieves the highest DS-F1 while maintaining or improving ID accuracy.

## 5 EXPERIMENTS

We conduct experiments on CIFAR-100 Krizhevsky et al. (2009) using ResNet-18 He et al. (2016) and on ImageNet-1K Deng et al. (2009) using DeiT-B Touvron et al. (2021), following the Near/Far OOD setup in the OpenOOD benchmark Yang et al. (2022); Zhang et al. (2023). Models are evaluated on ID test sets and Near/Far OOD datasets with both standard metrics (AURC, F1) and single scoring and our proposed double-scoring metrics (DS-AURC, DS-F1). For double scoring, we treat MSP as the ID score ($s_{\mathrm{ID}}$) and pair it with post-hoc OOD scores such as ODIN Liang et al. (2018) or EBO Liu et al. (2020); when MSP is used alone, the metrics reduce to the single-scoring case. Full details of the datasets and evaluation settings are provided in Appendix A.2. In addition, we report results under the standard OpenOOD protocol in Appendix A.4. These results highlight the OOD discriminative ability of post-hoc methods and confirm that our implementation is consistent with the official OpenOOD benchmark.

**Comparison between double scoring and single scoring.** We summarize the results of both the proposed double-scoring function and the single-scoring function in Table 1. For the single-scoring metrics, we report AURC and F1 on the test set containing both ID and OOD samples. For the double-scoring metrics, we evaluate our proposed DS-F1 and DS-AURC on the same test set. From the results, we highlight three key insights: *i) Double scoring consistently outperforms single scoring.* By combining two scores (a post-hoc method with MSP Hendrycks & Gimpel (2017)), our approach achieves significant and consistent improvements on both F1 and AURC, validating the effectiveness of DS-F1 and DS-AURC over single-score baselines. *ii) Far-OOD improvement is particularly notable.* Compared to the baseline MSP, double scoring delivers stronger gains on Far-OOD cases. This suggests that OOD methods are highly effective when handling distinct OOD samples, while their impact diminishes on the more challenging Near-OOD cases, which is consistent with the findings in OpenOOD Zhang et al. (2023). Although Near-OOD detection is generally regarded as more challenging, most existing methods are designed and evaluated on Far-OOD, leading to comparatively limited progress on Near-OOD. *iii) MSP remains a strong single-score baseline.* Even without double scoring, MSP demonstrates solid performance, especially on large-scale

| Metric: DS-AURC ↓ ResNet-18 - Trained on CIFAR-100 training set. Evaluated on CIFAR100 test set + Near-OOD. | | | | | | | | | | |
|---|---|---|---|---|---|---|---|---|---|---|
| **Training strategy** | MSP | OpenMax | MDS | Gram | ReAct | KLM | VIM | KNN | SIRC | Acc. |
| Basic | 202.38 | 201.78 | 201.78 | 201.14 | 199.61 | 199.23 | 198.39 | 197.30 | 197.08 | 77.32 |
| Mixup | 199.15 | 194.40 | 198.07 | 198.70 | 199.04 | 197.36 | 197.93 | 193.16 | 198.98 | 78.47 |
| RegMixup | 184.32 | 183.40 | 184.15 | 184.32 | 184.22 | 183.03 | 183.78 | 182.80 | 184.08 | 79.35 |
| AugMix | 212.08 | 211.71 | 211.78 | 209.38 | 204.51 | 211.52 | 206.50 | 206.42 | 202.19 | 76.98 |
| PixMix | 210.44 | 209.78 | 209.78 | 206.15 | 200.31 | 209.78 | 203.68 | 206.99 | 199.77 | 77.20 |
| CutMix | 257.72 | 248.02 | 246.43 | 256.78 | 256.78 | 246.96 | 252.25 | 216.56 | 256.78 | 77.81 |
| SURE | 199.05 | 191.80 | 198.78 | 196.41 | 198.78 | 198.11 | 198.78 | 182.86 | 193.09 | 80.55 |
| - CSC | 188.42 | 187.78 | 187.78 | 185.59 | 180.23 | 184.64 | 187.35 | 187.48 | 185.31 | 80.36 |
| - CRL | 187.34 | 186.78 | 186.78 | 186.60 | 182.15 | 186.78 | 185.66 | 185.43 | 183.83 | 80.68 |
| + SWA→EMA | 183.61 | 182.78 | 182.78 | 183.01 | 177.42 | 181.67 | 183.23 | 182.78 | 180.41 | 80.54 |
| + SAM→F-SAM | 181.86 | 180.78 | 180.78 | 179.79 | 178.19 | 180.73 | 181.47 | 180.82 | 178.58 | 80.79 |
| + RegPixMix (**SURE+**) | **173.45** | **173.13** | **172.78** | **171.51** | **169.02** | **171.06** | **172.95** | **173.45** | **170.28** | **81.66** |

| Metric: DS-AURC ↓ ResNet-18 - Trained on CIFAR-100 training set. Evaluated on CIFAR100 test set + Far-OOD. | | | | | | | | | | |
|---|---|---|---|---|---|---|---|---|---|---|
| Basic | 367.56 | 365.87 | 355.07 | 313.08 | 331.40 | 351.05 | 321.09 | 334.68 | 353.41 | 77.32 |
| Mixup | 357.54 | **308.17** | 345.87 | 351.96 | 352.68 | 344.72 | 346.39 | 319.28 | 356.78 | 78.47 |
| RegMixup | 398.83 | 364.21 | 385.03 | 364.54 | 384.43 | 382.76 | 388.94 | 339.67 | 397.89 | 79.35 |
| AugMix | 341.92 | 336.07 | 335.16 | 328.47 | 301.55 | 335.61 | 318.61 | 321.53 | 327.13 | 76.98 |
| PixMix | 374.21 | 373.78 | 361.44 | 305.26 | 322.75 | 344.67 | 328.09 | 324.76 | 336.33 | 77.20 |
| CutMix | 426.99 | 350.24 | 374.27 | 406.69 | 417.24 | 390.84 | 402.06 | 312.48 | 425.78 | 77.81 |
| SURE | 393.25 | 374.63 | 392.43 | 341.42 | 380.26 | 386.29 | 385.16 | 339.69 | 355.93 | 80.55 |
| - CSC | 367.92 | 335.35 | 347.37 | 351.69 | 299.64 | 335.85 | 302.61 | 298.00 | 361.24 | 80.36 |
| - CRL | 352.49 | 337.60 | 341.63 | 339.22 | 301.00 | 329.13 | **274.06** | 270.08 | 343.65 | 80.68 |
| + SWA→EMA | 355.63 | 343.26 | 346.65 | 340.78 | 299.51 | 337.65 | 305.04 | 273.98 | 345.06 | 80.54 |
| + SAM→F-SAM | 368.00 | 345.65 | 357.42 | 340.81 | 308.97 | 340.58 | 338.15 | 293.52 | 357.88 | 80.79 |
| + RegPixMix (**SURE+**) | **314.04** | 313.78 | **309.11** | **262.85** | **284.43** | **294.40** | 293.50 | **252.85** | **293.09** | **81.66** |

Table 3: **DS-AURC results on CIFAR-100 (ResNet-18) with different training strategies**. We report DS-AURC scores (lower is better) for both near- and far-OOD settings, along with ID accuracy. SURE+ consistently achieves the lowest DS-AURC while maintaining or improving ID accuracy.

datasets such as ImageNet Deng et al. (2009). This further emphasizes that while MSP is strong, double scoring pushes performance beyond its limits.

**Effectiveness of SURE+.** We evaluate the proposed SURE+ approach on CIFAR100 with ResNet-18, with DS-F1 results shown in Table 2 and DS-AURC results in Table 3. We also provide a comprehensive comparison of DS-AURC and accuracy across other training paradigms, including Baseline (standard cross-entropy loss), Mixup Thulasidasan et al. (2019), Pixmix Hendrycks et al. (2022), Augmix Hendrycks* et al. (2020), Cutmix Yun et al. (2019), RegMixup Pinto et al. (2022), and SURE Li et al. (2024b), following the OpenOOD benchmark Yang et al. (2022); Zhang et al. (2023). Finally, we conduct an ablation study on SURE+ to isolate the contribution of each component and assess their individual and combined benefits. From the results, we see that SURE+ emerges as the most effective strategy for building reliable classifiers. Compared to SURE and other training paradigms, SURE+ consistently delivers superior performance on both DS-F1 and DS-AURC, while also achieving the highest ID classification accuracy. Its advantages hold across different OOD detection methods, underscoring its robustness. For completeness, results on single-task settings (OOD detection and failure prediction) are provided in Appendix A.3.

## 6 CONCLUSION

In this work, we present a unified perspective on classifier reliability, emphasizing the complementary nature of out-of-distribution (OOD) detection and failure prediction. We show that evaluating these aspects in isolation is suboptimal and propose two complementary metrics, DS-F1 and DS-AURC, to jointly assess a model's ability to identify OOD inputs and anticipate its own errors. Extensive experiments on the OpenOOD benchmark validate the effectiveness of our unified evaluation framework, demonstrating its ability to distinguish classifiers that are truly robust and trustworthy. We further introduce SURE+, an improved, reliable classifier that integrates recent advances in both OOD detection and failure prediction. Empirical results confirm that SURE+ achieves better reliability across different post-hoc scores, highlighting the practical value of our approach. We believe that our framework and metrics offer a principled foundation for future research on trustworthy AI systems.

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

# A APPENDIX

## A.1 DETAILS ON PROPOSED EVALUATION METRICS

### A.1.1 IMPLEMENTATION FOR DS-F1 AND DS-AURC

Algorithm 1 summarizes the computation procedure for DS-F1 defined in Eq. 2. Given a model $m$ and corresponding scoring functions $s_{\text{ID}}$ and $s_{\text{OOD}}$, the algorithm evaluates the F1 score over a grid of ID and OOD thresholds. For each threshold pair $(\tau_{\text{ID}}, \tau_{\text{OOD}})$, it computes precision and recall based on correct ID classifications. The final DS-F1 score is the maximum F1 obtained over all threshold combinations.

---

**Algorithm 1:** Compute DS-F1 on the evaluation set with both ID and OOD samples

---

**Input:** Model $m$, scoring functions $s_{\text{ID}}, s_{\text{OOD}}$, evaluation set $\mathcal{D} = \mathcal{D}_{\text{ID}} \cup \mathcal{D}_{\text{OOD}}$, number of thresholds $T_{\text{grid}}$ per score

**Output:** DS-F1 score

Compute score arrays for all samples: $s_{\text{ID}}(x_i), s_{\text{OOD}}(x_i)$;
Initialize DS-F1 $\leftarrow 0$;
Obtain $T_{\text{grid}}$ quantile thresholds for each score: $\{\tau_{\text{ID},t}\}_{t=1}^{T_{\text{grid}}}$, $\{\tau_{\text{OOD},t}\}_{t=1}^{T_{\text{grid}}}$;
**for** $\tau_{ID}$ in $\{\tau_{ID,t}\}$ **do**
    **for** $\tau_{OOD}$ in $\{\tau_{OOD,t}\}$ **do**
        $\mathcal{A} \leftarrow \{i : s_{\text{ID}}(x_i) \geq \tau_{\text{ID}} \wedge s_{\text{OOD}}(x_i) \geq \tau_{\text{OOD}}\}$;
        $|A_{\text{ID}}| \leftarrow |\mathcal{A} \cap \mathcal{D}_{\text{ID}}|$;
        **if** $|\mathcal{A}| == 0$ **then**
            F1 $\leftarrow 0$
        **else**
            $\epsilon_{\text{ID}} \leftarrow \sum_{i \in \mathcal{A} \cap \mathcal{D}_{\text{ID}}} \mathbb{I}(m(x_i) \neq y_i)$;
            precision $\leftarrow (|A_{\text{ID}}| - \epsilon_{\text{ID}})/|\mathcal{A}|$;
            recall $\leftarrow (|A_{\text{ID}}| - \epsilon_{\text{ID}})/|\mathcal{D}_{\text{ID}}|$;
            F1 $\leftarrow 2 \cdot \text{precision} \cdot \text{recall}/(\text{precision} + \text{recall})$;
        DS-F1 $\leftarrow \max(\text{DS-F1}, \text{F1})$;

**return** DS-F1

---

Algorithm 2 summarizes the computation procedure for DS-AURC. Following Eqs. 3, 6, and 7, we first discretize coverage into $K$ intervals and enumerate thresholds to obtain a finite set of $(u, \text{risk})$ pairs. For each threshold (or threshold pair in the double scoring case), the algorithm calculates the accepted sample set, computes the corresponding selective risk, and records the coverage level. Finally, the risk values are aggregated within each coverage interval, and the minimization is applied for the aggregation when we consider DS-AURC. Finally, we integrate numerically to obtain the final AURC or DS-AURC score.

### A.1.2 CONSISTENCY WITH SINGLE-SCORING METRICS

We provide formal arguments showing that DS-F1 and DS-AURC strictly generalize their standard counterparts. The key observations rely only on the definitions of minimum and maximum over sets.

**Lemma 1 (Property of minima)** *For any function* $g : \mathcal{X} \to \mathbb{R}$ *and any* $x_0 \in \mathcal{X}$, *it holds that*

$$\min_{x \in \mathcal{X}} g(x) \leq g(x_0).$$

**Lemma 2 (Property of maxima under set inclusion)** *Let* $h : \mathcal{X} \to \mathbb{R}$ *and* $S_1 \subseteq S_2 \subseteq \mathcal{X}$. *Then*

$$\max_{x \in S_1} h(x) \leq \max_{x \in S_2} h(x).$$

Both lemmas are direct consequences of the definitions of minimum and maximum.

---

**Algorithm 2:** Compute (DS-)AURC on the evaluation set with both ID and OOD samples

---

**Input:** Model $m$, scoring functions $s_{\text{ID}}$, $s_{\text{OOD}}$ (single or both), evaluation set $\mathcal{D} = \mathcal{D}_{\text{ID}} \cup \mathcal{D}_{\text{OOD}}$,
        number thresholds $T_{\text{single}}$ (single) or $T_{\text{grid}}$ (per-score), coverage intervals $K$
**Output:** AURC or DS-AURC

---

Compute score arrays for all samples: $s_{\text{ID}}(x_i), s_{\text{OOD}}(x_i)$ (if single scoring, denoted as $s(x_i)$);
Set $N \leftarrow |\mathcal{D}|$;
Initialize list of pairs $\mathcal{P} \leftarrow \varnothing$;
**if** *single scoring* **then**
    Obtain $T_{\text{single}}$ quantile thresholds $\{\tau_t\}$ for $s$;
    **for** $\tau$ *in* $\{\tau_t\}$ **do**
        $\mathcal{A} \leftarrow \{i : s(x_i) \geq \tau\}$;
        $|A| \leftarrow |\mathcal{A}|; |A_{\text{ID}}| \leftarrow |\mathcal{A} \cap \mathcal{D}_{\text{ID}}|; |A_{\text{OOD}}| \leftarrow |\mathcal{A} \cap \mathcal{D}_{\text{OOD}}|$;
        **if** $|A| == 0$ **then**
            $\lfloor$ risk $\leftarrow 0$
        **else**
            $\epsilon_{\text{id}} \leftarrow \sum_{i \in \mathcal{A} \cap \mathcal{D}_{\text{ID}}} \mathbb{I}(m(x_i) \neq y_i)$;
            risk $\leftarrow (\epsilon_{\text{id}} + |A_{\text{OOD}}|)/|A|$;
        coverage $u \leftarrow |A_{\text{ID}}|/|\mathcal{D}_{\text{ID}}|$;
        Append $(u, \text{risk})$ to $\mathcal{P}$;
**else**
    Obtain $T_{\text{grid}}$ quantile thresholds for each score: $\{\tau_{\text{ID},t}\}_{t=1}^{T_{\text{grid}}}, \{\tau_{\text{OOD},t}\}_{t=1}^{T_{\text{grid}}}$;
    **for** *each pair* $(\tau_{ID}, \tau_{OOD})$ **do**
        $\mathcal{A} \leftarrow \{i : s_{\text{ID}}(x_i) \geq \tau_{\text{ID}} \wedge s_{\text{OOD}}(x_i) \geq \tau_{\text{OOD}}\}$;
        $|A| \leftarrow |\mathcal{A}|; |A_{\text{ID}}| \leftarrow |\mathcal{A} \cap \mathcal{D}_{\text{ID}}|; |A_{\text{OOD}}| \leftarrow |\mathcal{A} \cap \mathcal{D}_{\text{OOD}}|$;
        **if** $|A| == 0$ **then**
            $\lfloor$ risk $\leftarrow 0$
        **else**
            $\epsilon_{\text{id}} \leftarrow \sum_{i \in \mathcal{A} \cap \mathcal{D}_{\text{ID}}} \mathbb{I}(m(x_i) \neq y_i)$;
            risk $\leftarrow (\epsilon_{\text{id}} + |A_{\text{OOD}}|)/|A|$;
        coverage $u \leftarrow |A_{\text{ID}}|/|\mathcal{D}_{\text{ID}}|$;
        Append $(u, \text{risk})$ to bucket for coverage $u$ (or to $\mathcal{P}$);

Create K coverage intervals over [0,1] (e.g., $(0, \frac{1}{K}], (\frac{1}{K}, \frac{2}{K}], ..., (\frac{K-1}{K}, 1]$);
For each interval, collect all risk values whose u falls into the interval; if empty, interpolate
  from neighbors; otherwise aggregate by $\phi$ (e.g., $min(\cdot)$) defined in Eq. 7;
**return** computed AUC;

---

**DS-F1 dominates F1.** Let $f(\tau_{\text{OOD}}, \tau_{\text{ID}})$ denote the ds-F1 score obtained under thresholds $(\tau_{\text{OOD}}, \tau_{\text{ID}})$. Then, we can simplify Eq. 2 as:

$$\text{DS-F1} = \max_{\tau_{\text{OOD}}, \tau_{\text{ID}}} f(\tau_{\text{OOD}}, \tau_{\text{ID}})$$

If we consider a system with both ID and OOD samples evaluating with the classic F1 score and the previous pipeline, the results become:

$$\text{F1} = \max_{\tau_{\text{ID}}} f(\tau'_{\text{OOD}}, \tau_{\text{ID}})$$

where $\tau'_{\text{OOD}}$ is a fixed optimal threshold obtained by a certain post-hoc method. Then we have:

$$\text{F1} \leq \text{DS-F1}$$

and this inequality will also stand when we have a fixed $\tau_{\text{ID}}$.

**Proof A.1** *Define the feasible sets*

$$S_1 = \{(\tau'_{OOD}, \tau_{ID}) \mid \tau_{ID} \in [0,1]\}, \quad S_2 = \{(\tau_{OOD}, \tau_{ID}) \mid \tau_{OOD}, \tau_{ID} \in [0,1]\}.$$

*Clearly $S_1 \subseteq S_2$. By Lemma 2,*

$$\max_{(\tau_{OOD}, \tau_{ID}) \in S_1} f(\tau_{OOD}, \tau_{ID}) \leq \max_{(\tau_{OOD}, \tau_{ID}) \in S_2} f(\tau_{OOD}, \tau_{ID}).$$

*This is equivalent to F1 $\leq$ DS-F1.*

**DS-AURC dominates AURC.** When we calculate the regular AURC to filter both misclassified ID samples and accepted OOD samples from the whole acceptance set defined in Eq. 1, we can consider that OOD filtering is fixed, the acceptance set is restricted to $\mathcal{A}(\tau'_{\text{OOD}}, \tau_{\text{ID}})$, where $\tau'_{\text{OOD}}$ is a fixed threshold for OOD rejection. Accordingly, we then can rewrite Eq. 6 as

$$\text{SelectiveRisk}(\tau_{\text{ID}}) = \frac{\sum_{i \in \mathcal{D}_{\text{ID}}} Z_i \cdot \mathbb{I}(i \in \mathcal{A}(\tau'_{\text{OOD}}, \tau_{\text{ID}})) \; + \; |\mathcal{A}(\tau'_{\text{OOD}}, \tau_{\text{ID}}) \cap \mathcal{D}_{\text{OOD}}|}{|\mathcal{A}(\tau'_{\text{OOD}}, \tau_{\text{ID}})|}$$

Since the coverage in Eq. 3 is defined over all admissible acceptance sets, we have

$$\Big\{ x \mid x = \text{SelectiveRisk}(\tau_{\text{ID}}) \Big\} \subseteq \Big\{ x \mid x = \text{SelectiveRisk}(\tau_{\text{OOD}}, \tau_{\text{ID}}) \Big\}.$$

That is, the set of risks attainable under standard AURC is a strict subset of those attainable under DS-AURC. Consequently, if we use the minimum selective risk under a given coverage as the aggregation operator (cf. Eq. 7), DS-AURC necessarily achieves lower risk:

$$\text{DS-AURC}(m, s_{\text{OOD}}, s_{\text{ID}}) \; \leq \; \text{AURC}(m, s).$$

**Proof A.2** *Given a fix $\tau_{OOD}$, by Lemma 1 we have,*

$$\min_{\tau_{ID}} SelectiveRisk(\tau_{OOD}, \tau_{ID}) \;\; \geq \;\; \min_{\tau_{OOD}, \tau_{ID}} SelectiveRisk(\tau_{OOD}, \tau_{ID}).$$

*Taking expectation over coverage, it follows that*

$$DS\text{-}AURC(m, s_{OOD}, s_{ID}) \;\; \leq \;\; AURC(m, s).$$

The inequalities above are tight when the fixed thresholds $(\tau'_{\text{OOD}}, \tau_{\text{ID}})$ or $(\tau_{\text{OOD}}, \tau'_{\text{ID}})$ coincide with the global optima of the joint problem. Otherwise, the inequalities are strict. This shows that:

- DS-F1 never performs worse than the standard F1, and can be strictly higher if ID and OOD thresholds interact non-trivially.

- DS-AURC never performs worse (i.e., never larger) than AURC, and can be strictly smaller in practice.

Thus, double scoring is a strict generalization of the traditional single-scoring evaluation pipeline. In particular, by jointly considering two scores during evaluation, the system can simultaneously account for correct ID predictions, misclassified ID samples, and OOD samples, leading to both lower selective risk and higher F1 score. This demonstrates the practical advantage and improved expressiveness of the double scoring framework.

## A.2 DATASETS AND ARCHITECTURES

**Datasets.** We evaluate on CIFAR-100 Krizhevsky et al. (2009) and ImageNet-1K Deng et al. (2009), following the widely used OpenOOD benchmark Yang et al. (2022). For CIFAR-100, which contains 50k training and 10k test images over 100 classes, the near-OOD datasets include CIFAR-10 Krizhevsky et al. (2009) and TinyImageNet Le & Yang (2015) (with 2,502 overlapping images removed), while the far-OOD datasets include MNIST LeCun (1998), FashionMNIST Xiao et al. (2017), Texture Kylberg (2011), and Places365 Zhou et al. (2017) (with 1,305 overlaps removed). For ImageNet-1K, the near-OOD is built using Species Basart et al. (2022) (10k subset from 713k images), iNaturalist Huang & Li (2021) (10k), ImageNet-O Hendrycks et al. (2021b) (2k), and OpenImage-O Wang et al. (2022) (17k), all curated to exclude in-distribution classes; far-OOD includes Texture Kylberg (2011), MNIST LeCun (1998), and SVHN Netzer et al. (2011).

| ResNet-18 - Trained on CIFAR-100 | | | | |
| --- | --- | --- | --- | --- |
| **Training strategy** | **OOD Detection** | | **Failure Prediction** | **Acc. ↑** |
| | **AUROC ↑** | **FPR@95 ↓** | **AURC ↓** | |
| Basic | $80.82_{\pm0.05}$ / $79.31_{\pm0.99}$ | $54.34_{\pm0.60}$ / $56.64_{\pm2.33}$ | $61.64_{\pm1.46}$ | 77.32 |
| Mixup | $80.54_{\pm0.02}$ / $79.25_{\pm0.42}$ | $55.47_{\pm0.02}$ / $55.99_{\pm0.91}$ | $57.15_{\pm0.83}$ | 78.47 |
| RegMixup | $81.11_{\pm0.44}$ / $76.69_{\pm0.91}$ | $51.75_{\pm1.17}$ / $59.86_{\pm1.87}$ | $51.71_{\pm1.27}$ | 79.35 |
| AugMix | $79.66_{\pm0.02}$ / $79.47_{\pm0.22}$ | $56.12_{\pm0.24}$ / $54.26_{\pm0.47}$ | $60.45_{\pm0.18}$ | 76.98 |
| PixMix | $79.88_{\pm0.54}$ / $81.02_{\pm1.00}$ | $55.41_{\pm0.08}$ / $53.50_{\pm1.65}$ | $60.02_{\pm2.03}$ | 77.20 |
| CutMix | $77.66_{\pm1.49}$ / $76.49_{\pm1.87}$ | $68.40_{\pm6.15}$ / $67.40_{\pm6.76}$ | $68.68_{\pm7.06}$ | 77.81 |
| SURE | $79.23_{\pm0.22}$ / $74.24_{\pm0.46}$ | $58.69_{\pm0.69}$ / $63.91_{\pm0.33}$ | $44.59_{\pm1.18}$ | 80.55 |
| - CSC | $80.50_{\pm0.09}$ / $76.42_{\pm1.32}$ | $54.40_{\pm0.55}$ / $59.88_{\pm1.61}$ | $44.66_{\pm0.76}$ | 80.36 |
| - CRL | $80.51_{\pm0.16}$ / $77.40_{\pm0.34}$ | $54.04_{\pm0.51}$ / $56.59_{\pm0.98}$ | $43.68_{\pm1.34}$ | 80.68 |
| + SWA → EMA | $80.91_{\pm0.66}$ / $77.69_{\pm0.69}$ | $53.30_{\pm1.08}$ / $56.54_{\pm2.37}$ | $43.72_{\pm2.38}$ | 80.54 |
| + SAM → FSAM | $81.14_{\pm0.29}$ / $77.16_{\pm0.55}$ | $52.94_{\pm0.51}$ / $57.45_{\pm1.71}$ | $43.29_{\pm0.97}$ | 80.79 |
| + RegPixMix (**SURE+**) | $82.06_{\pm0.60}$ / $82.95_{\pm0.49}$ | $51.41_{\pm1.73}$ / $48.26_{\pm1.11}$ | $39.44_{\pm1.99}$ | 81.66 |

Table 4: Comparison of **OOD detection** metrics (AUROC, FPR@95) on the CIFAR-100 test set and its near/far-OOD datasets, **failure prediction** metric (AURC) on the CIFAR-100. All experiments are conducted with the MSP as the scoring function under different training strategies. Each result is averaged over three runs, and both the mean and standard deviation are reported. The top-3 methods for each metric are highlighted using a color gradient from light blue to **dark blue**.

**Archtectures.** Following the OpenOOD benchmark Yang et al. (2022), we conduct experiments on CIFAR-100 as the in-distribution (ID) dataset using ResNet-18 He et al. (2016) for small-scale evaluation. For large-scale experiments with ImageNet Deng et al. (2009) as the ID dataset, we adopt the transformer-based architecture DeiT-B Touvron et al. (2021). Experiments on CIFAR are repeated three times, and we report the average results.

**Training settings.** We unify the optimization settings across all methods, including a batch size of 128, SGD with momentum 0.9, an initial learning rate of 0.1 with cosine annealing Loshchilov & Hutter (2017), and a weight decay of 5e-3. Models are trained for 200 epochs, and results on CIFAR-100 are averaged over three independent runs to reduce stochastic variability.

**Evaluation settings.** We evaluate the models and post-hoc methods on the corresponding ID test sets, along with Near/Far OOD sets, reporting both standard metrics (AURC, F1) and the proposed metrics (DS-AURC, DS-F1). Different from OpenOOD Yang et al. (2022); Zhang et al. (2023), which only evaluates ID–OOD discrimination, SURE Li et al. (2024b)/FMFP Zhu et al. (2023b), which only considers ID test set failure prediction, and SIRC Xia & Bouganis (2022), which treats the two problems simultaneously but separately, our setting directly evaluates selective classification when both ID and OOD samples coexist. For the proposed double-scoring metrics, we adopt MSP Hendrycks & Gimpel (2017) as one scoring function, while the other is provided by different post-hoc methods, such as ODIN Liang et al. (2018), EBO Liu et al. (2020), etc. When the post-hoc method itself is MSP, our metrics naturally reduce to the single-scoring case.

For the single-scoring metrics, we simply use the score from each post-hoc method as the sole scoring measure. All hyperparameters of post-hoc methods are selected automatically using both ID and OOD validation samples, with the configuration maximizing AUROC used for final testing. Throughout this paper, unless otherwise noted, all metrics are multiplied by 100, while AURC is multiplied by 1000. For completeness, results under the OpenOOD protocol are provided in Appendix A.4, highlighting the OOD discriminative ability of post-hoc methods and confirming the consistency of our implementation with standard OpenOOD benchmarks.

## A.3 PERFORMANCE OF SURE+ ON SEPARATE TASKS

In the main paper, we primarily evaluate different training strategies using the proposed metrics, i.e., DS-F1 and DS-AURC. Here, we further provide a more detailed analysis by reporting the conventional metrics separately for OOD detection (AUROC, FPR), ID failure prediction (AURC), and Accuracy. As shown in Table 4, we observe that the model reliability benefits more from the com-

| Method | Model: ResNet-18 Training strategy: Basic Training/Validation set: CIFAR-100 training set Evaluation set: CIFAR-100 test set + Near/Far OOD sets | | |
|---|---|---|---|
| | AUROC↑ | FPR@95↓ | AUPR↑ |
| MSP Hendrycks & Gimpel (2017) | 80.82±0.05 / 79.31±0.99 | 54.34±0.60 / 56.64±2.33 | 83.99±2.33 / 66.73±1.71 |
| OpenMax Bendale & Boult (2016) | 72.34±0.37 / 73.10±0.81 | 55.56±0.21 / 59.83±1.63 | 80.10±1.63 / 62.92±1.38 |
| ODIN Liang et al. (2018) | 80.05±0.12 / 79.94±0.95 | 58.27±0.79 / 57.58±2.51 | 82.94±2.51 / 67.26±1.71 |
| MDS Lee et al. (2018) | 68.10±0.92 / 74.16±0.66 | 74.68±1.00 / 68.42±1.47 | 71.91±1.47 / 57.39±1.32 |
| Gram Hendrycks et al. (2021a) | 54.39±0.31 / 70.65±1.39 | 92.29±0.19 / 70.25±3.62 | 57.09±3.62 / 55.18±2.56 |
| EBO Liu et al. (2020) | 81.14±0.15 / 80.59±1.18 | 54.96±0.89 / 55.24±2.77 | 84.04±2.77 / 68.25±1.89 |
| GradNorm Huang et al. (2021) | 76.04±0.50 / 75.13±2.35 | 78.87±1.40 / 76.67±2.97 | 75.64±2.97 / 53.12±2.90 |
| ReAct Sun et al. (2021) | 80.71±0.43 / 81.45±1.11 | 56.73±2.29 / 52.05±2.70 | 83.31±2.70 / 69.41±2.02 |
| MLS Basart et al. (2022) | 81.21±0.12 / 80.40±1.07 | 54.88±0.83 / 55.33±2.71 | 84.07±2.71 / 68.08±1.83 |
| KLM Basart et al. (2022) | 74.88±0.46 / 73.32±0.81 | 78.69±4.95 / 82.01±2.23 | 75.05±2.23 / 50.89±0.91 |
| VIM Wang et al. (2022) | 73.43±0.07 / 79.14±0.99 | 63.98±0.42 / 54.93±0.84 | 77.84±0.84 / 67.34±0.81 |
| KNN Sun et al. (2022) | 80.51±0.12 / 80.93±0.28 | 59.31±0.87 / 57.32±1.65 | 81.73±1.65 / 67.56±1.43 |
| DICE Sun & Li (2022) | 80.34±0.26 / 80.60±1.40 | 55.97±1.03 / 55.03±3.26 | 83.52±3.26 / 68.35±2.18 |
| SIRC(MSP,$\|z\|_1$) Xia & Bouganis (2022) | 80.66±0.03 / 79.35±1.06 | 54.45±0.53 / 56.78±2.36 | 83.94±2.36 / 66.68±1.72 |
| SIRC(MSP,Res.) Xia & Bouganis (2022) | 80.75±0.05 / 79.11±1.12 | 54.41±0.57 / 56.68±2.35 | 83.97±2.35 / 66.68±1.75 |
| SIRC(-H,$\|z\|_1$) Xia & Bouganis (2022) | 81.10±0.13 / 80.17±1.12 | 54.50±0.56 / 56.37±2.47 | 84.20±2.47 / 67.39±1.79 |
| SIRC(-H,Res.) Xia & Bouganis (2022) | 81.07±0.11 / 79.80±1.15 | 54.43±0.65 / 56.27±2.43 | 84.18±2.43 / 67.35±1.81 |
| ID Acc. | 77.32 | | |

Table 5: **Evaluation on ResNet-18 based on OpenOOD settings.** We report each metric on both Near- and Far-OOD tests. All experiments were run three times, and both the average and the standard deviation are presented. The top-5 methods for each metric are highlighted using a color gradient from light blue to **dark blue**.

bination of different training strategies. Specifically, our proposed **SURE+** consistently achieves competitive or superior results across both tasks, demonstrating its effectiveness in improving model reliability from both the OOD Detection and failure prediction perspectives.

### A.4 FULL RESULTS BASED ON OPENOOD SETTINGS

We present full results based on OpenOOD settings Yang et al. (2022); Zhang et al. (2023) in this section. Unlike the double-scoring selective classification setup in the main paper experiments, where the scoring function was required to separate correctly classified ID samples from both misclassified ID and OOD samples, the OpenOOD setting follows the standard OOD detection protocol, focusing solely on distinguishing ID from ID+OOD data during evaluation. This evaluation is thus simpler but widely adopted in the OOD detection literature. Presenting these results aligns our evaluation with the established OpenOOD benchmark, verifies the correctness of our implementation, and complements the more challenging double-scoring selective classification experiments reported in the main paper.

Concretely, we report results on ResNet-18 He et al. (2016) trained on CIFAR-100, benchmarking a variety of post-hoc OOD detection methods within the unified OpenOOD framework. While we use the same model as in the main paper, the evaluation setting here differs: we adopt AUROC, FPR@95, and AUPR as metrics, following the OpenOOD protocol. Accordingly, the objective is also aligned with OpenOOD, focusing solely on distinguishing ID data from OOD data.

As shown in Table 5, we observe that the ranking of the results generally aligns with the results in the more recent OpenOOD 1.5 benchmark Zhang et al. (2023), which demonstrates the reliability of our implementation.

### A.5 IMPLEMENTATION DETAILS OF SURE+

SURE+ builds upon SURE Li et al. (2024b) with several modifications to its components. Specifically, the original SURE loss function consists of a standard cross-entropy (CE) loss, a RegMixup loss Pinto et al. (2022), and a CRL loss Dong et al. (2017), with a cosine-similarity classifier (CSC) as the prediction head. As described in the main paper, SURE+ removes the CRL loss and re-

places the CSC with the linear classifier and introduces an additional RegPixMix loss based on PixMix Hendrycks et al. (2022). Furthermore, SURE+ also applies F-SAM Li et al. (2024a) and EMA to replace FMFP optimizations Zhu et al. (2023b), i.e. SAM Foret et al. (2021) and SWA Izmailov et al. (2018). We list the implementation details as follows.

**RegMixup.** Given a training sample $(x, y)$ and a randomly shuffled pair $(x', y')$ from the same batch, we sample a mixing coefficient $\kappa \sim \text{Beta}(\alpha, \alpha)$ and construct a mixed input

$$\tilde{x} = \kappa x + (1 - \kappa)x',$$

The RegMixup loss is defined as a weighted sum of classification losses on the original and shuffled labels:

$$\mathcal{L}_{\text{RegMixup}} = \kappa \, \mathcal{L}_{\text{CE}}(m(\tilde{x}), y) + (1 - \kappa) \, \mathcal{L}_{\text{CE}}(m(\tilde{x}), y').$$

where $m(\cdot)$ denotes the network align with the notation in the main paper and $\mathcal{L}_{\text{CE}}(\cdot, \cdot)$ is the cross-entropy loss. This formulation is equivalent to training on a soft label $\kappa y + (1 - \kappa)y'$, and serves as a regularization term that improves robustness. Specifically, we follow the original RegMixup setting and adopt $\alpha = 10$ in the experiments.

**RegPixMix.** PixMix augments each input by stochastically mixing it with an auxiliary image $x_{\text{mix}}$ sampled from a set of texture images Hendrycks et al. (2022). Starting from the original image $x$, we optionally apply a random augmentation, and then repeatedly (1–4 times) select either another augmentation of $x$ or $x_{\text{mix}}$ to serve as the mixing partner. A mixing operator $M$ is then drawn from a predefined set of transformations $\mathcal{M}$, and the final mixed image is constructed as

$$x_{\text{pixmix}} = M(x, x_{\text{mix}}),$$

The corresponding loss is simply

$$\mathcal{L}_{\text{RegPixMix}} = \mathcal{L}_{\text{CE}}(m(x_{\text{pixmix}}), y).$$

**F-SAM.** To further enhance generalization, we employ *Friendly Sharpness-Aware Minimization (F-SAM)* Li et al. (2024a) as an optimization strategy. F-SAM perturbs the model parameters along the normalized gradient direction

$$\epsilon_{\text{FSAM}} = \rho_f \cdot \frac{\nabla_\theta \mathcal{L}}{\|\nabla_\theta \mathcal{L}\|_2},$$

and minimizes a convex combination of the standard and perturbed losses:

$$\mathcal{L}_{\text{FSAM}} = \gamma \, \mathcal{L}(\theta) + (1 - \gamma) \, \mathcal{L}(\theta + \epsilon_{\text{FSAM}}).$$

This formulation encourages convergence towards flatter minima while mitigating the over-regularization observed in standard SAM Foret et al. (2021).

**EMA.** We further stabilize training and evaluation with two complementary techniques. First, we maintain an *Exponential Moving Average (EMA)* of the model parameters:

$$\theta_{\text{EMA}} \leftarrow \tau \theta_{\text{EMA}} + (1 - \tau)\theta,$$

where $\tau$ is a momentum coefficient. The EMA parameters $\theta_{\text{EMA}}$ are used for evaluation, providing a smoother trajectory and reducing the variance of predictions. Second, following standard practice, we apply *Revisiting Batch Normalization (ReBN)* after the final training epoch, which updates the batch normalization statistics by running a single forward pass over the entire training set without gradient updates. EMA and ReBN jointly improve the stability of learned representations under distribution shifts, leading to more reliable calibration and robustness.

**Overall Objective.** The overall training objective of SURE+ is defined as

$$\mathcal{L}_{\text{SURE+}} = \mathcal{L}_{\text{CE}}(x, y) + \lambda_1 \mathcal{L}_{\text{RegMixup}} + \lambda_2 \mathcal{L}_{\text{RegPixMix}},$$

where we set $\lambda_1 = \lambda_2 = 1.0$ for all our experiments. The optimization is further regularized by F-SAM, and evaluation is performed with EMA (ReBN), leading to improved robustness, generalization, and calibration.

A.6 Use of LLMs

Large language models (LLMs) were used solely for writing refinement and did not participate in any part of the research process, including data analysis, retrieval, or discovery.

