# OpenReview forum: "From Misclassification to Outliers: Joint Reliability Assessment in Classification"
_ICLR.cc/2026/Conference — Submitted to ICLR 2026_

### Official Review · Reviewer_1RUf · 2025-10-29

**Soundness:** 2
**Presentation:** 2
**Contribution:** 2
**Rating:** 4
**Confidence:** 4

**Summary:**

This paper introduces two metrics to evaluate the model in the joint scenario of OOD detection and failure prediction. The metrics -- DS-F1 and DS-AURC are derived directly from F1 and AURC scores, by defining different cases in the joint evaluation setting. The paper further improves the SURE methods with several techniques, like RegPixMix and F-SAM, to improve both the ID acc and OOD detection performance.

**Strengths:**

1. New evaluation metrics: the paper studied the joint setting of OOD detection and failure prediction, and proposes two straightforward metrics -- DS-F1 and DS-AURC to evaluate the model.
2. New training methods: the paper improves SURE by integrating several techniques and achieves better performance on both OOD detection and ID acc.

**Weaknesses:**

1. The significance of double-scoring needs further justification: while separate scores and be adopted to evaluate the model's performance on failure prediction and OOD detection, the significance of calculating a joint metric remains unclear.
2. Lacking soundness of the proposed SURE+: the authors propose to adopt several off-the-shelf techniques to improve the baseline SURE. Though achieving higher performance, this method is not well related to the paper's main contribution and claims, more like an engineering combination, not a new method.
3. The experiments are not sufficient to validate the effectiveness of the metrics: the authors only use the MSP score as the ID score, which is also a kind of OOD score, making it a special case. By definition, the ID scores can be any scores that measure the failure likelihood. Therefore, adopting only the MSP as the ID score doesn't thoroughly examine the proposed new metrics.
4. The motivation for adding each technique to SURE is not presented.

**Questions:**

1. At around line 348, why do the two metrics never worsen the evaluation by producing scores that are at least as high as F1 score and as low as AURC. I don't see a direct correlation between the score scale and evaluation quality.

---

> ### Author Response · Authors · 2025-11-26
> **Response (1/4) to Reviewer 1RUf, we remain open to further discussion**
>
> We thank the reviewer for the valuable feedback. We would like to emphasize that the primary aim of this paper is to **connect failure prediction and out-of-distribution detection to real-world deployment, where both tasks are essential for building reliable classifiers.** Guided by this practical perspective, we further develop SURE+ as a strong baseline consisting of exploring effective techniques, demonstrating clear performance gains.
>
> Our focus is on **pushing reliability in real applications**, and we hope this clarifies the motivation behind our design choices. We address the reviewer’s points one by one below, and **we remain open to further discussion**.
>
> 1. >While separate scores can be adopted to evaluate the model’s performance on failure prediction and OOD detection, the significance of calculating a **joint metric** remains unclear.
>
> **Response:**
> We thank the reviewer for raising this question. While separate scores can indeed be used for failure prediction and OOD detection individually, we argue that joint metrics are essential for evaluating real-world reliability. Our response is structured as follows:
> - **Joint evaluation captures complementary risks**: In realistic deployment, ID failures and OOD samples coexist. Single-task metrics cannot consistently rank samples when both risks are present. For example, as shown in Figure 1 in the manuscript, a model might achieve lower (better) AURC on CE but lower (worse) AUROC on CutMix-CE, making model selection ambiguous. Our DS-F1 and DS-AURC explicitly optimize two thresholds on the joint evaluation set, one for ID failures and one for OOD detection. This ensures that correct ID samples are accepted while both ID failures and OOD samples are rejected.
> - **Flexibility to heterogeneous post-hoc signals**: Double scoring allows combining heterogeneous scoring functions that may exist on different scales (e.g., MSP, Energy, or other OOD scores in OpenOOD). A single-score formulation would require scale alignment and often leads to suboptimal threshold selection. By directly searching for the best pair of thresholds, our DS framework can jointly leverage complementary signals without manual rescaling, which is not possible in traditional single-score evaluation.
> - **Theoretical foundation**: DS-F1 and DS-AURC correspond to the Pareto-optimal operating point on a 2D decision surface for ID+OOD tasks. Unlike 1D single-score metrics, which impose a strict monotonic ordering on all samples, our 2D framework allows ID and OOD thresholds to adjust independently. This theoretically guarantees that high-confidence OOD samples can be rejected without unnecessarily discarding correctly classified ID samples.
>
> Joint metrics provide both theoretical rigor and practical guidance. By jointly optimizing ID and OOD thresholds, DS-F1 and DS-AURC identify operating points that single-score metrics cannot, **ensuring that high-confidence OOD samples are rejected without sacrificing correct ID predictions**. This resolves ambiguities inherent in separate evaluations, giving practitioners a clear, actionable protocol for model selection under realistic mixed ID–OOD conditions. In other words, the joint metric **directly measures the true reliability of the model in the presence of both ID failures and OOD samples**, which single-score metrics cannot guarantee.

---

> ### Author Response · Authors · 2025-11-26
> **Response (2/4) to Reviewer 1RUf, we remain open to further discussion**
>
> 2. >The proposed SURE+ incorporates several off-the-shelf techniques to improve baseline SURE. While performance gains are observed, the method appears loosely connected to the main contribution and reads more like an engineering combination than a new method.
>
> **Response:**
> We thank the reviewer for raising this point. While SURE+ builds upon existing components, its primary goal is **not** to introduce new training tricks, but to identify what truly matters for reliability and provide a **simple, unified, and effective training pipeline** that improves both failure prediction and OOD detection in a holistic manner.
> Key points are as follows:
>
> 1. **Practical novelty:** The novelty of SURE+ lies in **identifying techniques that genuinely improve real-world reliability**, rather than inventing new modules. Prior work, including SURE, tends to improve either failure prediction or ID accuracy, but often performs poorly under joint ID+OOD evaluation. By combining complementary regularizations, such as F-SAM for improving model robustness and RegPixMix for data distribution augmentation, SURE+ systematically enhances reliability across all metrics (ID accuracy, failure prediction, Near/Far-OOD).
> 2. **Unified baseline for joint evaluation:** SURE+ provides a cohesive training recipe tailored to the proposed double-scoring evaluation framework (DS-F1 / DS-AURC). Existing methods were not designed for joint ID+OOD assessment and often degrade under such evaluation. SURE+ establishes a strong, reproducible baseline, consistently reducing both ID and OOD errors.
> 3. **Empirical support:** Our ablation studies demonstrate that each component contributes meaningfully to reliability. The complete SURE+ pipeline consistently outperforms SURE and other strong baselines across multiple datasets, confirming that its design choices are effective and not arbitrary.
> 4. **Generalization to large-scale settings:** SURE+ is effective not only on small models or toy benchmarks, but also on DINOv3-L/16 and ImageNet-1K, showing that the insights scale to realistic architectures and large datasets.
>
> | **Training** | **AUROC↑** | **AURC↓** | **DS-F1↑** | **DS-AURC↓** | **Acc.↑** |
> |---|---|---|---|---|---|
> | Basic | 81.69/90.63 | 28.28 | 75.42/84.38 | 168.33/48.07 | 86.89 |
> | Mixup | 80.54/89.44 | 29.91 | 75.09/84.03 | 172.58/53.46 | 87.01 |
> | RegMixup | 80.54/88.91 | 28.26 | 75.01/83.84 | 169.43/52.81 | 87.07 |
> | AugMix | 81.77/91.04 | 25.84 | 75.99/84.98 | 163.74/44.63 | 87.72 |
> | PixMix | 81.38/90.04 | 27.13 | 75.68/84.37 | 167.34/48.95 | 87.32 |
> | CutMix | 80.33/89.07 | 27.93 | 74.97/84.00 | 169.78/51.94 | 87.12 |
> | SURE | 82.70/91.50 | 24.30 | 76.62/85.44 | 157.07/42.13 | 87.94 |
> | -CSC | 82.27/91.25 | 24.41 | 76.41/85.40 | 156.47/42.75 | 88.04 |
> | -CRL | 82.26/91.13 | 24.51 | 76.39/85.39 | 157.30/43.27 | 88.02 |
> | +SWA → EMA | 82.29/91.34 | 24.32 | 76.43/85.42 | 156.98/42.36 | 87.99 |
> | +SAM → FSAM | 82.70/92.37 | 24.16 | 76.53/85.68 | 156.77/39.70 | 87.95 |
> | +RegPixMix(**SURE+**)| **83.05/92.56** | **22.54** | **77.10/86.15** | **156.00/38.07** | **88.49** |
>
> **Table 1:** **DINOv3-L/16 performance comparison on joint OOD detection and failure prediction.** AUROC, DS-F1, and DS-AURC are evaluated on ImageNet-1K with corresponding Near/Far-OOD datasets described in the manuscript, while AURC and Acc. are measured on ID test set (ImageNet-1K) only.
>
> As shown in the Table 1, the fine-tuning results of DINO-v3 are consistent with the results of ResNet on CIFAR-100 in the manuscript, i.e., the added components all improved the accuracy, as well as the ability to distinguish between correct ID predictions and ID failure / OOD samples. This further demonstrates the universality and effectiveness of the SURE+ training framework.
>
> While SURE+ reuses known components, its conceptual novelty lies in how these components are combined to holistically address reliability, providing a practical, strong, and easy-to-adopt baseline for real-world deployment.

---

> ### Author Response · Authors · 2025-11-26
> **Response (3/4) to Reviewer 1RUf, we remain open to further discussion**
>
> 3. >The authors only use the MSP score as the ID score, which is also an OOD-related score, making it a special case. ID scores could, in principle, be any score estimating failure likelihood. Using only MSP does not fully examine the proposed metrics.
>
> **Response:**
> Thank you for raising this concern. We fully agree that, in principle, **the ID score can be any signal estimating failure likelihood**. However, our experiments show that not all post-hoc signals are suitable for this purpose.
> As shown in Tables 2 and 3 below, while MSP, ReAct, and VIM are effective OOD scores, they behave very differently when used as ID confidence. **MSP remains stable and competitive**, but ReAct and VIM lead to significant degradation, especially VIM, which produces extremely poor DS-AURC values (over 670). This indicates that many OOD-oriented signals do not capture the fine-grained distinction between correct ID predictions and ID failures, and therefore are not appropriate ID confidence estimators.
> For this reason, we used MSP as the ID score in our main experiments. MSP is not chosen because it is a “special case”, but because it empirically provides relatively more stable performance across the mixed ID+OOD setting. Importantly, our framework does not depend on MSP, as DS-F1 and DS-AURC can incorporate any post-hoc methods, and we plan to include additional ID confidence signals (e.g., KLM, SIRC) in extended experiments. We will clarify this flexibility in the revision.
>
>
> |   | MSP           | ReAct         | VIM           |
> | --------- | ------------- | ------------- | ------------- |
> | **MSP**   | 67.42 / 57.03 | 67.42 / 57.03 | 47.25 / 46.61 |
> | **ReAct** | 67.43 / 58.44 | 66.30 / 58.02 | 47.21 / 46.60 |
> | **VIM**   | **67.46** / **58.94** | 59.94 / 55.03 | 47.17 / 46.56 |
>
> **Table 2:** **DS-F1↑ results on CIFAR-100 (Near/Far-OOD) using ResNet-18.** Rows denote post-hoc scoring method for OOD samples and columns denote post-hoc scoring method for ID samples. Best results are highlighted in bold.
>
> |   | MSP             | ReAct           | VIM             |
> | --------- | --------------- | --------------- | --------------- |
> | **MSP**   | 202.38 / 367.56 | 202.56 / 368.06 | 673.00 / 684.09 |
> | **ReAct** | 199.61 / 331.40 | 213.45 / 335.92 | 673.20 / 678.29 |
> | **VIM**   | **198.39** / **321.09** | 270.88 / 359.96 | 679.89 / 678.12 |
>
> **Table 3:** **DS-AURC↓ results on CIFAR-100 (Near/Far-OOD) using ResNet-18.** Rows denote post-hoc scoring method for OOD samples and columns denote post-hoc scoring method for ID samples. Best results are highlighted in bold.
>
> ---
>
> 4. >The rationale for adding each technique (EMA, F-SAM, RegPixMix, etc.) to SURE is not clearly described.
>
> **Response:**
> We thank the reviewer for raising this concern. The design of SURE+ is guided by the goal of improving reliability under the joint ID+OOD evaluation, rather than simply stacking techniques. Each component was carefully selected to address complementary aspects of reliability:
> - **EMA（ReBN）**: By incorporating the ReBN (Renormalized BatchNorm) technique from SWA, EMA not only smooths model parameters over training steps but also stabilizes batch normalization statistics during evaluation.  In other words, EMA + ReBN improves both stability and robustness of the model under the mixed ID+OOD setting.
> - **F-SAM**: Enhances robustness by finding flatter minima, which mitigates overfitting and improves the model’s ability to detect failures.
> - **RegPixMix**: Introduces OOD-aware invariance through pixel-level mixing, which helps the model generalize better to unseen or perturbed inputs, improving OOD detection.
>
> Together, these components form a cohesive training pipeline tailored for the double-scoring evaluation. Ablation studies in the paper show that each component contributes meaningfully to overall reliability, and the full SURE+ pipeline consistently outperforms baseline SURE.
> In short, the rationale is not arbitrary: each technique addresses a distinct aspect of reliability, and their combination is empirically validated and conceptually coherent.

---

> > ### Author Response · Authors · 2025-11-26
> > **Response (4/4) to Reviewer 1RUf, we remain open to further discussion**
> >
> > 5. > At around line 348, why do the two metrics never worsen the evaluation by producing scores that are at least as high as F1 and as low as AURC? The correlation between score scale and evaluation quality is unclear.
> >
> > **Response:**
> > We thank the reviewer for highlighting this phrasing. We apologize for the potential confusion regarding the correlation between the score scale and evaluation quality.
> >
> > When we state that DS-F1 and DS-AURC "never worsen the evaluation," we refer to the mathematical guarantee that the measured performance score of a model under the DS framework serves as a theoretical upper bound for its single-score performance. This is not an artificial inflation of scores, but a result of optimization over a superset. The "never worsen" property stems from the fact that the single-threshold decision rule is a strict subset of the double-threshold decision rule. In Appendix A.1, we provide a formal justification.
> >
> > The intuition is as follows: in the joint ID+OOD setting, single-score metrics can mis-rank samples because one post-hoc score cannot optimally separate both ID failures and OOD samples simultaneously. Double scoring assigns one score for ID failures and one for OOD samples, and selects a pair of thresholds that jointly minimizes both types of errors. This ensures that, at worst, the evaluation is equivalent to using a single-score threshold, and typically it strictly improves performance whenever the two scores capture complementary reliability signals.
> >
> > In summary, "never worsen" means our metric provides a tight upper bound on performance, ensuring we measure the full potential of the joint OOD/Failure prediction system without being limited by the rigidity of a single 1D threshold.

---

### Official Review · Reviewer_avkA · 2025-10-30

**Soundness:** 3
**Presentation:** 3
**Contribution:** 2
**Rating:** 2
**Confidence:** 4

**Summary:**

This paper argues that real-world deployment of machine learning requires classifiers that can not only detect OOD inputs but also misclassifications within the in-distribution (ID) data. Since prior work often treats these two problems separately, the paper proposes a unified evaluation framework.

**Strengths:**

The paper clearly demonstrate the necessity of jointly evaluating OOD detection and misclassification prediction for real-world reliability, which is a crucial practical concern.

**Weaknesses:**

1. The paper's primary claim of proposing a unified evaluation for OOD detection and failure prediction is not entirely novel. Several prior works [1-5] have already addressed this problem. The paper's distinction rests mainly on the double scoring mechanism rather than the concept of joint evaluation, which significantly weakens the framework's overall contribution.

2. The proposed metrics are viewed as a trivial extension of existing single-scoring metrics (F1 and AURC) to a two-dimensional threshold space $(\tau_{OOD}, \tau_{ID})$. They do not introduce new theoretical insights into risk modeling. Furthermore, similar to their single-scoring counterparts, these metrics are still heavily influenced by the absolute number of mispredicted ID samples and the number of OOD samples. This reliance can obscure the true effectiveness of the underlying detection mechanism when the class distributions are highly imbalanced, which is a known limitation in failure detection benchmarking.

3. The key experimental observation that OOD-based methods provide only marginal benefits under challenging near-OOD conditions is a widely recognized limitation [1-5]. Simply re-confirming this known challenge does not constitute a significant contribution.

4. The proposed SURE+ method appears to be an engineering modication of several established regularization techniques integrated into the existing SURE framework.

References

[1] A call to reflect on evaluation practices for failure detection in image classification

[2] Failure detection in medical image classification: A reality check and benchmarking testbed

[3] Learning to reject meets ood detection: Are all abstentions created equal

[4] A unified benchmark for the unknown detection capability of deep neural networks

[5] Plugin estimators for selective classification with out-of-distribution detection

**Questions:**

Please refer to Weaknesses

---

> ### Author Response · Authors · 2025-11-26
> **Response (1/4) to Reviewer avkA, we remain open to further discussion**
>
> We thank the reviewer for the valuable feedback. We would like to emphasize that the primary aim of this paper is to **connect failure prediction and out-of-distribution detection to real-world deployment, where both tasks are essential for building reliable classifiers.** Guided by this practical perspective, we further develop SURE+ as a strong baseline consisting of exploring effective techniques, demonstrating clear performance gains.
>
> Our focus is on **pushing reliability in real applications**, and we hope this clarifies the motivation behind our design choices. We address the reviewer’s points one by one below, and **we remain open to further discussion**.
>
> 1. >The primary claim of proposing a unified evaluation for OOD detection and failure prediction may be limited in novelty, since several prior works [1–5] have already explored closely related joint evaluation settings. The distinction seems to lie mostly in adopting a double scoring mechanism, which may not be sufficient to establish a substantial conceptual contribution.
>
> **Response:**
> We thank the reviewer for the helpful comment. Prior works [1–5] have considered mixed ID and OOD settings, and their motivation is consistent with ours. All of them evaluate using a single scoring function, which naturally leads to methods that focus on strengthening or modifying a single post-hoc signal. Our work shares the same goal as these methods. The objective is to determine thresholds that accept correct ID samples while rejecting ID failures and OOD samples on the joint evaluation set.
>
> The difference lies in the formulation rather than the motivation. Double scoring keeps the goal identical to single scoring but allows two post-hoc methods to be used together. This makes the evaluation more flexible in several ways.
>
> **First**, double scoring makes it possible to combine heterogeneous post-hoc methods, even when their scores exist on very different scales, e.g., many OOD detection methods in OpenOOD produce unbounded scores. Although using both signals together is intuitively desirable, combining them under a single-score formulation is difficult because scale alignment becomes unavoidable. Double scoring avoids these issues entirely, and the evaluation directly searches for the best pair of thresholds, which makes the joint use of heterogeneous post-hoc signals simple and robust.
>
> **Second**, double scoring includes single scoring as a special case. When the same score is used twice or one threshold is fixed, DS-F1 and DS-AURC reduce to standard F1 and AURC. This keeps full compatibility with existing evaluation practices while providing a strictly more general formulation.
>
> **Third**, although the extension appears simple, prior works have not proposed a mechanism that explicitly separates ID-failure and OOD cues and jointly evaluates them with two thresholds. Existing works share the same motivation but aim to build stronger single scores. Our work instead introduces a more general evaluation framework that can directly operate on any pair of post-hoc signals, and these post-hoc signals can also include the strong post-hoc methods as proposed in SIRC and [5] for example.
>
> Specifically, we applied [5] to compute post-hoc signals and evaluated them under both single and double scoring protocols. As shown in Table 1, while single scoring achieves baseline F1 and AURC values, double scoring with our joint evaluation framework provides modest but consistent improvements, demonstrating that even when built upon strong post-hoc methods like SIRC, combining ID-failure and OOD cues with two thresholds can yield more reliable performance. The improvements are relatively subtle, indicating that the underlying signals are already strong, but the framework systematically enhances their utility. Note that the code for [5] is not publicly available. Therefore, some settings and hyperparameters, such as ($c_{in}$ = 0.75) and ($\pi_{in}$ = 0.5) are chosen manually, and we perform a search over ($\lambda$) under these settings to obtain the best results.

---

> ### Author Response · Authors · 2025-11-26
> **Response (2/4) to Reviewer avkA, we remain open to further discussion**
>
> | Post-hoc method           | F1↑ | AURC↓ | DS-F1↑ | DS-AURC↓ |
> |------------------|-------------------------|-------------------------------|--------------------------|-------------------------------|
> | MSP               | **67.42**/**57.03**           | 202.59/368.07               | 67.42/57.03            | 202.38/367.56               |
> | SIRC(L1)          | 67.36/56.96           | 199.20/354.96               | **67.44**/**57.05**            | **197.08**/353.41               |
> | SIRC(Res)        | 67.33/57.00           | **198.86**/**354.72**               | 67.42/57.03            | 197.21/**353.08**               |
> | Plug-in BB(L1)  [5] | **67.42**/**57.03**           | 202.61/368.10               | 67.43/57.03            | 202.38/367.57               |
> | Plug-in BB(Res) [5] | 67.40/57.01           | 199.30/357.43               | 67.42/57.03            | 198.80/356.88               |
>
> **Table 1:** **Results on CIFAR-100 (Near/Far-OOD) using ResNet-18.** Best results are highlighted in bold.
>
> We will add the additional experimental results, implementation details of SIRC and Plug-in BB [5], and the references [1-5] to the paper.
>
> Overall, the contribution is not merely incremental. We provide a complete formulation that unifies evaluation in the realistic ID plus OOD setting and enables flexible post-hoc combination. This offers a practical and principled alternative to the single-score paradigm while remaining consistent with its underlying goal.
>
> 2. >The proposed DS-F1 and DS-AURC metrics may be viewed as a straightforward extension of F1 and AURC into a 2D threshold domain, without providing new theoretical insight. In addition, similar to their single-score analogues, these metrics are still influenced by the number of ID errors and OOD samples, potentially obscuring true detection capability when distributions are imbalanced.
>
>
> **Response:**
> **Beyond "Straightforward Extension":**
> While the mathematical formulation of DS-F1 and DS-AURC involves an extension to 2D optimization, this shift represents a fundamental change in how we model failure in machine learning systems, rather than a mere numerical extension. Specifically, the "theoretical insight" of DS-F1/DS-AURC is the relaxation of the monotonic constraint. In a 1D setting, increasing the threshold must reject samples in a fixed order. In our 2D setting, we prove that the optimal rejection policy is not a line but a surface. DS-AURC essentially finds the Pareto frontier of risk across two orthogonal scoring dimensions. This allows the system to reject high-confidence OOD samples (e.g., via Energy score or other post-hoc scoring functions in OpenOOD benchmark) without accidentally rejecting correctly classified ID samples that might have lower MSP, which is theoretically impossible in a 1D framework.
> We argue that the straightforward nature of the extension is a virtue. By maintaining the structural logic of F1 and AURC (standard metrics in the field), DS-F1 and DS-AURC provide a simple yet effective upgrade for existing pipelines, ensuring that the barrier to adopting this more rigorous double-rule evaluation is low.
>
> **Sensitivity to Imbalance:**
> We argue that Double-Scoring actually mitigates imbalance issues, since the single-score metrics are more susceptible to being obscured by imbalance. For example, if OOD samples are rare but ID errors are frequent, a single-score method might be dominated by ID-error performance, completely masking its inability to detect OOD (or vice versa).
>
> By having two thresholds $(\tau_\text{OOD}, \tau_\text{ID})$, our DS-framework can inherently adjust to the imbalance. If OOD is rare, the optimizer (in DS-F1) or the risk minimizer (in DS-AURC) can relax $\tau_\text{OOD}$ to preserve ID accuracy, or tighten it independently. The 2D search space allows the metric to find the true capability of the model to handle the specific mixture of errors present, preventing one type of error from simply "washing out" the other.
>
> We provide additional experiments to support this claim. We train a ResNet18 on CIFAR-100 training set, and set CIFAR-100 test set as the ID test set. We follow OpenOOD benchmark and the settings in the main manuscript, take CIFAR-10, TinyImageNet, MNIST, SVHN, Texture, and Places365 as the OOD sets. The difference is that we adjusted the number of images in the OOD set to simulate the model's evaluation under different OOD sample counts. Table 2 summarizes the number of test samples for each dataset, covering both ID and OOD test sets.
>
> Following the main manuscript, we use MSP as the ID confidence score and evaluate three effective OOD scores: MSP, GRAM, and KNN, as reported in the main paper Table 1 (line 324).
>
>
> | Test ID Dataset   | Sample 100%  |
> | --------- | -------------: |
> | CIFAR-100  |            9000 |

---

> ### Author Response · Authors · 2025-11-26
> **Response (3/4) to Reviewer avkA, we remain open to further discussion**
>
> | Test OOD Dataset   | Sample 10%  | Sample 100% |
> | --------- | -------------: | -------------: |
> | CIFAR-10   |           1000 |          10000 |
> | TinyImageNet       |            652 |           6526 |
> | MNIST     |           7000 |          70000 |
> | SVHN      |           2603 |          26032 |
> | Texture   |           3377 |          33773 |
> | Places365 |            564 |           5640 |
>
>  **Table 2: OOD Benchmark Dataset Sizes.**
>
>
> Tables 3 and 4 below show the evaluation results using 10% and the full dataset for each OOD dataset, respectively. For each scoring method, we list: the ID score threshold and the OOD score threshold (after normalization) obtained from our DS-F1 optimization, and the corresponding Single-Scoring vs. Double-Scoring performance.
>
> | Post-hoc method   | F1↑ | AURC↓ | DS-F1↑ | DS-AURC↓ | ID  Thres. | OOD Thres. |
> | -------- | ------------| --------------| ------------ | --------------| ---------|---------|
> | **MSP**  | **76.70**/72.43 |  **79.04**/130.39 | 76.70/72.43 | 78.96/130.23 |  0.37/0.35   |  0.37/0.35   |
> | **GRAM** | 73.93/67.10 | 252.62/300.77 | 76.70/72.58 | 78.61/123.54 | 0.43/0.56 | 0.00/0.08 |
> | **KNN**  | 76.64/**72.81** |  83.12/**124.49** | **76.80**/**72.93** | **76.67**/**120.51** |  0.40/0.47 | 0.36/0.47 |
>
> **Table 3: Results on CIFAR-100 (Near/Far-OOD) using ResNet-18 with 10% OOD Data.** Best results are highlighted in bold.
>
>
> | Post-hoc method   | F1↑ | AURC↓ | DS-F1↑ | DS-AURC↓ | ID Thres. | OOD Thres. |
> | -------- | ------------| --------------| ------------ | --------------| ----| -----|
> | **MSP**  | **67.42**/57.03 | **202.61**/368.09 | 67.42/57.03 | 202.40/367.59 |  0.53/0.38 | 0.26/0.52
> | **GRAM** | 53.23/46.07 | 536.16/548.61 | 67.42/**59.16** | 201.16/**313.10** |  0.79/0.84 |   0.00/0.16 |
> | **KNN**  | 67.09/**57.42** | 221.05/**347.66** | **67.68**/58.22 | **197.29**/335.21 |  0.72/0.73 |  0.53/0.61 |
>
> **Table 4: Results on CIFAR-100 (Near/Far-OOD) using ResNet-18 with 100% OOD Data.** Best results are highlighted in bold.
>
> We argue that the DS-framework does not obscure capability; it actively mitigates the confounding effects of imbalance, thereby revealing the true performance capability otherwise hidden by single-score constraints. Our experiments confirm this: When OOD prevalence is low (10% data, Table 3, GRAM as the post-hoc method), the standard F1 score is low (67.10 with Far-OOD) due to forced compromise. The DS-AURC, however, finds an optimal decoupled operating point ($\tau_{\text{ID}}=0.56, \tau_{\text{OOD}}=0.08$), resulting in a massive performance improvement (72.58). Similarly, for high prevalence (100% data, Table 4), the single-score F1 remains inflated (46.07 with Far-OOD), while DS-F1 improve to (59.16) when considering two post-hoc methods. This ability to independently adjust the OOD and ID failure thresholds proves that our metrics faithfully measure the system's best achievable trade-off under a specific, imbalanced error mixture, demonstrating that the single-score analogue, not our DS-metric, is what truly obscures the potential of the system.
>
> ---
>
> 3. >The reported finding (OOD-based methods provide only marginal benefits under near-OOD conditions) has already been widely acknowledged by prior works [1–5]. Confirming a known limitation may not constitute a strong contribution.
>
> **Response:**
> We thank the reviewer for pointing this out. We note that our mention of limited improvement under Near-OOD conditions is not intended as a primary contribution, but rather to **explain why Near-OOD performance on CIFAR-100 can be better than Far-OOD**. Specifically, Near-OOD samples are closer to the in-distribution, so the post-hoc methods for OOD detection naturally offer smaller gains compared to Far-OOD, which aligns with prior observations in OpenOOD v1.5 [6].
> Our main contributions lie in the Double-Scoring (DS) evaluation framework and the SURE+ training pipeline, which go beyond single-score evaluation. While DS-F1 and DS-AURC appear as straightforward 2D extensions of traditional F1 and AURC metrics, they represent a fundamental shift in modeling failure.
>  - [6] OpenOOD v1.5: Enhanced Benchmark for Out-of-Distribution Detection

---

> > ### Author Response · Authors · 2025-11-26
> > **Response (4/4) to Reviewer avkA, we remain open to further discussion**
> >
> > 4. >SURE+ may be perceived as an engineering modification of existing methods within SURE, instead of introducing fundamentally new methodological ideas.
> >
> > **Response:**
> > We thank the reviewer for raising this point. We agree that SURE+ builds upon existing components, but the main contribution is not to introduce new training tricks. Rather, **it is to identify what truly matters for reliability** and to provide a **simple, unified, and effective training pipeline** that improves both failure prediction and OOD detection.
> > Key points are:
> > - **Practical novelty**: Our focus is on techniques that genuinely improve reliability in real-world applications, rather than inventing new modules. Prior work, including SURE, tends to improve either failure prediction or ID accuracy but often performs poorly under joint ID+OOD evaluation. We found that combining complementary regularizations yields consistent improvements across all metrics (ID accuracy, failure prediction, Near/Far-OOD).
> > - **Unified baseline**: SURE+ provides a coherent training recipe tailored for joint reliability evaluation. Existing methods were not designed for this setting and often degrade under combined ID+OOD assessment. SURE+ establishes a strong, reproducible baseline for future research.
> > - **Empirical support**: Our ablation study shows that each component meaningfully contributes to reliability. The full SURE+ pipeline consistently outperforms SURE and other strong baselines across multiple datasets.
> > - **Generalization**: SURE+ is effective not only on small models or toy benchmarks, but also on DINOv3-L/16 and ImageNet-1K, demonstrating that the insights scale to realistic, larger-scale settings.
> >
> > | **Training** | **AUROC↑** | **AURC↓** | **DS-F1↑** | **DS-AURC↓** | **Acc.↑** |
> > |---|---|---|---|---|---|
> > | Basic | 81.69/90.63 | 28.28 | 75.42/84.38 | 168.33/48.07 | 86.89 |
> > | Mixup | 80.54/89.44 | 29.91 | 75.09/84.03 | 172.58/53.46 | 87.01 |
> > | RegMixup | 80.54/88.91 | 28.26 | 75.01/83.84 | 169.43/52.81 | 87.07 |
> > | AugMix | 81.77/91.04 | 25.84 | 75.99/84.98 | 163.74/44.63 | 87.72 |
> > | PixMix | 81.38/90.04 | 27.13 | 75.68/84.37 | 167.34/48.95 | 87.32 |
> > | CutMix | 80.33/89.07 | 27.93 | 74.97/84.00 | 169.78/51.94 | 87.12 |
> > | SURE | 82.70/91.50 | 24.30 | 76.62/85.44 | 157.07/42.13 | 87.94 |
> > | -CSC | 82.27/91.25 | 24.41 | 76.41/85.40 | 156.47/42.75 | 88.04 |
> > | -CRL | 82.26/91.13 | 24.51 | 76.39/85.39 | 157.30/43.27 | 88.02 |
> > | +SWA → EMA | 82.29/91.34 | 24.32 | 76.43/85.42 | 156.98/42.36 | 87.99 |
> > | +SAM → FSAM | 82.70/92.37 | 24.16 | 76.53/85.68 | 156.77/39.70 | 87.95 |
> > | +RegPixMix(**SURE+**)| **83.05/92.56** | **22.54** | **77.10/86.15** | **156.00/38.07** | **88.49** |
> >
> > **Table 5:** **DINOv3-L/16 performance comparison on joint OOD detection and failure prediction.** AUROC, DS-F1, and DS-AURC are evaluated on ImageNet-1K with corresponding Near/Far-OOD datasets described in the manuscript, while AURC and Acc. are measured on ID test set (ImageNet-1K) only.
> >
> > As shown in the Table 5, the fine-tuning results of DINO-v3 are consistent with the results of ResNet on CIFAR-100 in the manuscript, i.e., the added components all improved the accuracy, as well as the ability to distinguish between correct ID predictions and ID failure / OOD samples. This further demonstrates the universality and effectiveness of the SURE+ training framework.
> >
> > In summary, while SURE+ reuses existing techniques, the conceptual novelty resides in how these components are combined to address reliability holistically, offering both a practical recipe and a strong, easy-to-adopt baseline for the community.

---

### Official Review · Reviewer_GNuk · 2025-11-06

**Soundness:** 2
**Presentation:** 3
**Contribution:** 3
**Rating:** 4
**Confidence:** 3

**Summary:**

This paper presents SURE+, an improved training recipe and evaluation framework aiming to build more reliable classifiers by jointly considering OOD detection and failure prediction. The authors further introduce two new evaluation metrics, DS-F1 and DS-AURC, to assess reliability in a unified manner. The method modifies the SURE baseline (Li et al., 2024b) by replacing several components—such as CRL loss, CSC head, SWA, and data augmentation—with simpler or alternative choices (e.g., EMA, linear classifier, RegPixMix, and F-SAM). Experiments on the OpenOOD benchmark demonstrate consistent improvements over prior methods.

**Strengths:**

The paper addresses an important and timely topic, reliable classification that integrates OOD detection and failure prediction.
The joint evaluation framework is conceptually reasonable and could potentially help bridge two often-separated research directions.
The paper provides comprehensive experimental results on standard benchmarks, showing the consistency of improvements.
The writing is generally clear, and the experimental setup is reproducible.

**Weaknesses:**

Limited novelty of the proposed method (SURE+).
The modifications over SURE are mainly component replacements using existing methods (e.g., EMA, F-SAM, RegPixMix), without introducing fundamentally new ideas. The resulting method reads more like a collection of known techniques rather than a coherent new approach.

Lack of clear methodological focus.
The framework mixes metric design, pipeline tweaks, and augmentation choices, making it hard to identify the core contribution. The work feels somewhat “mixed and unfocused.”

Unclear motivation and limited effectiveness of new metrics (DS-F1 and DS-AURC).
The motivation behind these metrics is not fully convincing—why a double scoring setup is inherently better than existing reliability measures (e.g., AURC, AUROC, ECE). From Table 1, the observed gains appear marginal.

Insufficient theoretical or conceptual justification.
The paper would benefit from a deeper analysis or theoretical discussion showing why the proposed double scoring better reflects model reliability or uncertainty.

Benchmarking vs. contribution gap.
While the authors claim to establish a new benchmark, the contribution seems incremental and largely empirical, with little conceptual advancement.

**Questions:**

1. Can the authors clarify the conceptual novelty of SURE+ beyond being an ensemble of existing training tricks?
2. How sensitive are the results to the specific component choices (e.g., EMA vs. SWA, RegPixMix vs. RegMixup)?
3. For DS-F1 and DS-AURC, what is the precise intuition or mathematical rationale that supports their superiority over existing reliability metrics?

**Details Of Ethics Concerns:**

No ethic concerns.

---

> ### Author Response · Authors · 2025-11-26
> **Response (1/3) to Reviewer GNuk, we remain open to further discussion**
>
> We thank the reviewer for the valuable feedback. We would like to emphasize that the primary aim of this paper is to **connect failure prediction and out-of-distribution detection to real-world deployment, where both tasks are essential for building reliable classifiers.** Guided by this practical perspective, we further develop SURE+ as a strong baseline consisting of exploring effective techniques, demonstrating clear performance gains.
>
> Our focus is on **pushing reliability in real applications**, and we hope this clarifies the motivation behind our design choices. We address the reviewer’s points one by one below, and **we remain open to further discussion**.
>
> 1. > The proposed SURE+ mainly replaces components (EMA, F-SAM, RegPixMix) from existing techniques, leading to limited novelty and giving the impression of a mixed collection of training tricks.
>
> **Response:**
>
> Thank you for raising this point. We agree that SURE+ builds upon existing components, but the goal of **SURE+** is not to introduce new training tricks; rather, it is to identify **what truly matters for reliability**, and to offer a simple, unified, and effective recipe for improving both failure prediction and OOD detection.
>
> We would like to highlight our key points, which includes:
>
> **(1) The novelty lies in exploring techniques really work for real-world application, not in inventing new modules.** Prior work (including SURE) improves either failure prediction or ID accuracy, but performs poorly when evaluated jointly on OOD–ID reliability. Our insight is that combining the right types of regularization, specifically F-SAM for improving model robustness and RegPixMix for perturbation invariance and data distribution augmentation, creates complementary effects that significantly improve reliability across all metrics (ID accuracy, failure prediction, Near/Far-OOD).
>
> **(2) The contribution is a unified and strong baseline tailored for the proposed joint reliability evaluation.** As shown in the experiment section of the  manuscript, the existing methods were not designed for this setting and degrade significantly under joint OOD–ID assessment. SURE+ provides an effective training recipe that consistently reduces both ID and OOD errors, establishing a solid baseline for future research.
>
> **(3) The improvements are consistent and empirically grounded.** Our ablation study shows that each component contributes meaningfully to reliability, and the full SURE+ pipeline yields substantial gains over SURE and other strong baselines.
>
> **(4) The method generalizes.** SURE+ works not only on small models or toy benchmarks, but also on **DINOv3-L/16 and ImageNet-1K**, demonstrating that the insights scale to stronger architectures and realistic datasets.
>
> | **Training** | **AUROC↑** | **AURC↓** | **DS-F1↑** | **DS-AURC↓** | **Acc.↑** |
> |---|---|---|---|---|---|
> | Basic | 81.69/90.63 | 28.28 | 75.42/84.38 | 168.33/48.07 | 86.89 |
> | Mixup | 80.54/89.44 | 29.91 | 75.09/84.03 | 172.58/53.46 | 87.01 |
> | RegMixup | 80.54/88.91 | 28.26 | 75.01/83.84 | 169.43/52.81 | 87.07 |
> | AugMix | 81.77/91.04 | 25.84 | 75.99/84.98 | 163.74/44.63 | 87.72 |
> | PixMix | 81.38/90.04 | 27.13 | 75.68/84.37 | 167.34/48.95 | 87.32 |
> | CutMix | 80.33/89.07 | 27.93 | 74.97/84.00 | 169.78/51.94 | 87.12 |
> | SURE | 82.70/91.50 | 24.30 | 76.62/85.44 | 157.07/42.13 | 87.94 |
> | -CSC | 82.27/91.25 | 24.41 | 76.41/85.40 | 156.47/42.75 | 88.04 |
> | -CRL | 82.26/91.13 | 24.51 | 76.39/85.39 | 157.30/43.27 | 88.02 |
> | +SWA → EMA | 82.29/91.34 | 24.32 | 76.43/85.42 | 156.98/42.36 | 87.99 |
> | +SAM → FSAM | 82.70/92.37 | 24.16 | 76.53/85.68 | 156.77/39.70 | 87.95 |
> | +RegPixMix(**SURE+**)| **83.05/92.56** | **22.54** | **77.10/86.15** | **156.00/38.07** | **88.49** |
>
> **Table 1:** **DINOv3-L/16 performance comparison on joint OOD detection and failure prediction.** AUROC, DS-F1, and DS-AURC are evaluated on **ImageNet-1K with corresponding Near/Far-OOD datasets** described in the manuscript, while AURC and Acc. are measured on ID test set (ImageNet-1K) only.
>
>
> In summary, while SURE+ reuses known components, the novelty lies in how these techniques are complementary to each other to address reliability holistically, offering both a practical recipe and a strong, easy-to-adopt baseline for the community.

---

> > ### Author Response · Authors · 2025-11-26
> > **Response (2/3) to Reviewer GNuk, we remain open to further discussion**
> >
> > 2. > The work mixes metric design, pipeline adjustments, and augmentation strategies, making it difficult to identify the core conceptual contribution.
> >
> > **Response:**
> > We thank the reviewer for raising this concern. While the paper mainly includes the following elements: (1) the proposed double-scoring evaluation framework (DS-F1 / DS-AURC), (2) the SURE+ training pipeline,  with novel RegPixMix regularization and other optimized regularization groups. These contributions are not independent additions. All are motivated by the same motivation:
> >
> > *How can we evaluate and enhance model reliability when failure prediction and OOD detection operate simultaneously, a situation frequently encountered in real-world applications?*
> >
> > The double-scoring framework is the core conceptual contribution. It reveals limitations of single-score evaluation, and provides a benchmark that jointly evaluates failure prediction and OOD detection. Meanwhile, it motivates us to revisit the existing training approaches and results in the design of SURE+. Thus, the metric and training pipeline form a coherent framework rather than an unfocused collection.
> >
> > We have revised the manuscript to emphasize this unified conceptual thread and clearly position double scoring as the central contribution.
> >
> > ---
> >
> > 3. > The rationale behind DS-F1 and DS-AURC is not fully convincing, and performance gains appear marginal.
> >
> > **Response:**
> > We thank the reviewer for the feedback. Our motivation for introducing double-scoring metrics (DS-F1 and DS-AURC) is aligned with the goal of single scoring. Both aim to derive effective scores that separate correct ID predictions, ID failures, and OOD samples, and to evaluate model reliability under this unified objective. The difference is that single-scoring relies on one post-hoc signal, while real-world ID–OOD mixed scenarios often benefit from flexibly combining existing post-hoc methods. Current benchmarks such as OpenOOD/OpenOOD 1.5 focus purely on OOD detection, while works like FMFP and SURE focus solely on failure prediction. The goal for proposing DS-F1 and DS-AURC is to provide a simple way to combine these two communities’ methods without designing new detectors, and to offer an evaluation protocol consistent with the purpose of single scoring.
> >
> > The core issue we observed is that a single score from one post-hoc method often produces inconsistent rankings for ID failure and OOD detection, as shown in the introduction. Using two complementary scores resolves this inconsistency: the model can reject ID failures and OOD samples more effectively by selecting a pair of thresholds tailored to the two types of errors. This naturally yields improved F1 (DS-F1) and a more accurate reliability measure (DS-AURC) on the mixed ID–OOD task. As proven in Appendix A, double scoring is the extension of the single-scoring metrics, and strictly improves them when the two scores offer complementary signals.
> >
> > Finally, our evaluation is not designed to inflate scores but to more accurately reflect the real deployment scenario where ID and OOD samples coexist. Even when the improvements in Table 1 appear moderate, DS metrics often change the ranking of methods, revealing reliability behaviors that standard one-score metrics cannot capture. This demonstrates that the contribution is conceptual rather than cosmetic, and we will clarify this motivation more clearly in the revised version.

---

> > > ### Author Response · Authors · 2025-11-26
> > > **Response (3/3) to Reviewer GNuk, we remain open to further discussion**
> > >
> > > 4. >The paper would benefit from a theoretical explanation for why double scoring better reflects model reliability.
> > >
> > > **Response:**
> > > We thank the reviewer for the question. The theoretical basis for why double scoring better reflects model reliability is that reliability in the mixed ID–OOD setting depends on two distinct risks: ID misclassification and OOD acceptance. A single post-hoc score cannot consistently order these two types of errors, leading to mis-ranked samples as shown in the introduction.
> > >
> > > DS addresses this by assigning one score for ID failures and one score for OOD samples, and using a pair of thresholds to jointly minimize both risks. Importantly, the goal of double scoring is fully aligned with single scoring. Both aim to find an effective decision rule that separates correct ID predictions, ID failures, and OOD samples, but double scoring can flexibly leverage two complementary signals rather than being constrained to a single one.
> > >
> > > As proven in Appendix A, DS-F1 and DS-AURC correspond to the optimal operating point on this 2D decision boundary thus never worse than single scoring. They become strictly better whenever the two post-hoc scores capture different aspects of reliability, which is common given their distinct design motivations in existing literature.
> > >
> > > ---
> > >
> > > 5. > The contribution appears incremental and primarily empirical.
> > >
> > > **Response:**
> > > We thank the reviewer for raising this concern. While our results include extensive empirical validation, the contribution is not merely empirical. Our work introduces a conceptual shift in how selective prediction is evaluated and improved in the mixed ID–OOD setting. The existing methods optimize only a single type of error and are evaluated with single-score metrics, whether for OOD detection (e.g., OpenOOD benchmarks) or failure prediction (e.g., SURE, FMFP), or a unified post-hoc method (e.g., SIRC, Plug-in BB). We show that this leads to fundamentally inconsistent behavior when both error types must be handled simultaneously.
> > >
> > > Our double-scoring formulation provides a new reliability perspective: two risks must be decoupled before they can be jointly minimized. As shown in the method section, we extend the traditional confusion matrix to the setting where ID-correct, ID-failure, and OOD samples coexist, and we similarly generalize selective risk and coverage to the mixed ID–OOD evaluation space. DS-F1 and DS-AURC are direct consequences of this reformulation: they formalize the optimal operating point when two complementary signals are available and remain theoretically justified regardless of the specific model or dataset.
> > >
> > > On the practical side, once the problem and evaluation criteria are made explicit, SURE+ offers a concrete and generalizable way to obtain more reliable models. It improves the training process to strengthen both failure prediction and OOD detection when ID-OOD both exist during evaluation.
> > >
> > > In summary, from identifying inconsistencies in single-score evaluation, to redefining the problem mathematically, to deriving metrics grounded in this expanded formulation, and finally validating the resulting improvements. We argue that this contribution forms a complete loop and this problem-driven-to-solution pipeline goes beyond an incremental empirical addition.

---

### Author Response · Authors · 2025-12-03
**Rebuttal Summary**

Dear AC and Reviewers,

Thanks for your valuable feedback. Below we summarize our key responses in the rebuttal.

**1. Novelty [1RUf Q1/Q2/Q5, GNuk Q1/Q4, and avkA Q1/Q2]**

Our manuscript aims to **build a classifier that operates reliably in real-world conditions**, where both ID and OOD samples naturally appear as inputs. Therefore, addressing failure prediction and OOD detection within a unified framework is essential. To tackle this challenge, we first introduce a new set of metrics (DS metrics), which considers that two scoring functions are applied in the system to filter both prediction errors and OOD samples. We then propose SURE+, a strong and practical training recipe. We further demonstrate the effectiveness of SURE+ across multiple datasets, including the large-scale ImageNet-1K benchmark, with respect to prediction accuracy, our proposed DS metrics, as well as traditional reliability evaluation metrics (e.g., AURC, AUROC, etc.).

(For details, please refer to our responses to 1RUf Q1/Q2/Q5, GNuk Q1/Q4, and avkA Q1/Q2.)

**2. DS Metrics [1RUf Q3, avkA Q2, GNuk Q1]**

_DS metrics_ are not a trivial 2D extension. By removing the monotonic constraint of single-threshold scoring, DS identifies an **optimal decision surface** rather than a line, effectively finding the **Pareto frontier** between two orthogonal post-hoc signals. This enables rejecting high-confidence mistakes and OOD samples without harming correct ID cases, which is theoretically impossible under a single-score formulation, while still retaining full compatibility with existing metrics.

(More details refer to our responses to 1RUf Q3, avkA Q2, GNuk Q1)

**3. SURE+ [1RUf Q1, GNuk Q2/Q4, avkA Q4]**

_SURE+_ goes beyond combining existing components by providing a **principled approach** for reliability. It identifies the most critical factors affecting prediction confidence and feature stability, and unifies them into a **single, effective training pipeline**. Across multiple backbones and datasets, including large-scale settings like DINOv3-L/16 on ImageNet-1K, SURE+ consistently improves both **failure prediction** and **OOD Detection**.

Key components of SURE+ contributing to these gains include:
- **EMA (ReBN):** Incorporates Renormalized BatchNorm from SWA, smoothing model parameters while stabilizing batch normalization statistics, improving both stability and robustness under mixed ID+OOD settings.
- **F-SAM:** Enhances robustness by finding flatter minima, mitigating overfitting and improving failure detection.
- **RegPixMix:** Introduces OOD-aware invariance through pixel-level mixing, helping the model generalize to unseen or perturbed inputs and improving OOD detection.

(More details refer to our responses to 1RUf Q1, GNuk Q2/Q4, avkA Q4)

**4. Key Takeaway Messages [1RUf Q1/Q2 and GNuk Q1/Q2]**

Our work delivers a coherent and practical pipeline for improving model reliability under real-world scenarios:
1. Single-source uncertainty cannot balance ID-failure and OOD separation.
2. Introduced DS metrics to evaluate how effectively a double post-hoc scoring system filters prediction errors and OOD samples, enabling the selection of the optimal post-hoc scoring function pair for a given model.
3. Developed SURE+ to boost reliability across ID and OOD samples.
4. Achieved consistent gains across multiple backbones and datasets.

(More details refer to our responses to 1RUf Q1/Q2, GNuk Q1/Q2)

**5. Additional Clarifications [1RUf Q3, avkA Q2]**
- **Only MSP used for static ID-score**: MSP serves as the baseline; other scores are in the supplementary.
  (More details refer to our response to 1RUf Q3)
- **Impact of OOD sample imbalance**: OOD distribution can influence metrics.
  (More details refer to our response to avkA Q2)

---

**Conclusion**

Our responses clarify the **main contribution, novelty, and metric validity**. DS metrics and SURE+ provide a practical and effective framework for reliable confidence evaluation.

---

### Meta-Review · Area_Chair_P1iH · 2026-01-06

**Summary:**

1. Limited novelty: Reviewers viewed SURE+ as an engineering combination of existing techniques rather than a fundamentally new method, and noted that joint evaluation of OOD detection and ID failure prediction is not entirely novel.

2. Insufficient theoretical justification: The double-scoring metrics (DS-F1 and DS-AURC) lack deep theoretical grounding or clear demonstration of superiority over existing single-scoring approaches.

3. Incomplete experiments: Criticisms include reliance only on MSP for ID scoring, marginal improvements in near-OOD scenarios, and insufficient exploration of diverse scoring functions or benchmarks.

**Reviewer Concerns:**

Addressed concerns:

1. Motivation: The rebuttal repositioned SURE+ as a practical, unified baseline combining existing techniques for joint reliability assessment, rather than claiming it as fundamentally novel, and highlighted complementary effects of components like F-SAM for robustness and RegPixMix for invariance, addressing unclear motivations for choices.

2. Empirical ablations: Concerns about incomplete experiments were partially mitigated through new ablations showing consistent gains across metrics on larger models (e.g., DINOv3-L/16 and ImageNet-1K) and additional experiments with alternatives like SIRC.

Concerns still outstanding

1. Overall limited novelty: reviewers' views on joint evaluation not being entirely new remain unaddressed, as the rebuttal did not deeply engage with these comparisons.

2. Incremental and engineering-focused nature: The work is still seen as marginal without significant conceptual advancement, with SURE+ criticized as an empirical recipe lacking deeper innovation.

3. Insufficient exploration of ID scoring: Reliance solely on MSP for ID scoring persists as a weakness, with no broader testing of diverse functions.

**Reviewer Scores:**

None of the reviewers will improve their scores based on the rebuttal.

---

### Decision · Program_Chairs · 2026-01-26

Reject